# Rejuvenating Cross-Entropy Loss in Knowledge Distillation for Recommender Systems

**Zhangchi Zhu**[1], **Wei Zhang**[1,2*]
[1]East China Normal University    [2]Shanghai Innovation Institute
zczhu@stu.ecnu.edu.cn, zhangwei.thu2011@gmail.com

## Abstract

This paper analyzes Cross-Entropy (CE) loss in knowledge distillation (KD) for recommender systems. KD for recommender systems targets at distilling rankings, especially among items most likely to be preferred, and can only be computed on a small subset of items. Considering these features, we reveal the connection between CE loss and NDCG in the field of KD. We prove that when performing KD on an item subset, minimizing CE loss maximizes the lower bound of NDCG, only if an assumption of closure is satisfied. It requires that the item subset consists of the student's top items. However, this contradicts our goal of distilling rankings of the teacher's top items. We empirically demonstrate the vast gap between these two kinds of top items. To bridge the gap between our goal and theoretical support, we propose **R**ejuvenated **C**ross-**E**ntropy for **K**nowledge **D**istillation (RCE-KD). It splits the top items given by the teacher into two subsets based on whether they are highly ranked by the student. For the subset that defies the condition, a sampling strategy is devised to use teacher-student collaboration to approximate our assumption of closure. We also combine the losses on the two subsets adaptively. Extensive experiments demonstrate the effectiveness of our method. Our code is available at https://github.com/BDML-lab/RCE-KD.

## 1 Introduction

Recently, as the scaling law in recommender systems (Zhai et al., 2024) has been increasingly recognized, many researchers have proposed extremely large models (Ohsaka & Togashi, 2023; Zhai et al., 2024) to achieve higher recommendation accuracy. However, the increase in model size inevitably incurs high storage costs and inference latency, causing higher maintenance costs and lower user satisfaction.

To improve the inference efficiency and decrease the storage cost of recommendation models without sacrificing their recommendation accuracy, knowledge distillation (KD) for recommender systems (Kang et al., 2020; Sun et al., 2024) has attracted attention. KD (Hinton et al., 2015) is an approach for model compression. It aims to transfer knowledge from a pre-trained large teacher to a small student. Once training is complete, only the small student is used for inference. Among existing works on KD, response-based KD (Hinton et al., 2015) encourages students to mimic the teacher's predictions and has gained extreme attention due to its excellent performance. As a popular loss for response-based KD methods, Cross-Entropy (CE) loss is very important. Most response-based KD methods (Huang et al., 2022; Cui et al., 2023) in Computer Vision (CV) and Natural Language Processing (NLP) are based on CE loss. However, little work has been done to use or analyze CE loss in KD for recommender systems. Note that KD for recommender systems has two unique features: 1) It focuses more on rankings than specific scores, especially among the teacher's top items (Kang et al., 2020). 2) KD can only be conducted on a small subset of items since the quantity of all the items is very large. These features make the compatibility of CE loss in KD for recommender systems questionable. To obtain an initial insight into the performance of CE loss, we present the results of vanilla CE loss and several response-based KD methods in Figure 1. To cover as many types of loss functions as possible, we consider the point-wise loss (i.e., CD (Lee et al., 2019)), pair-wise loss (i.e., UnKD (Chen et al., 2023)), and RRD-based losses (Kang et al., 2020) (a list-wise loss, i.e.,

---

*Corresponding author.

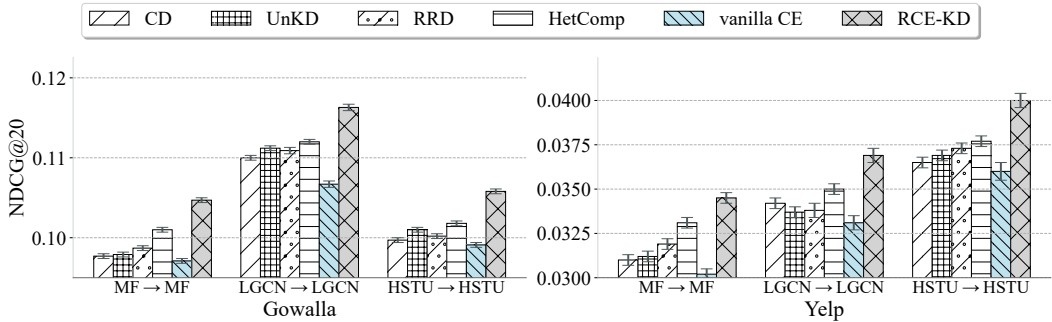

Figure 1: Performance comparison of different KD methods. We report the results in three homogeneous Teacher → Student settings.

RRD (Kang et al., 2020), HetComp (Kang et al., 2023)), and our method RCE-KD. In vanilla CE loss, we compute CE loss using the teacher's top items. We find that vanilla CE loss is often inferior to all baselines. This result contrasts with the extensive use of CE loss for KD in other fields.

Considering the features of recommender systems and the surprisingly poor performance of CE Loss, we analyze CE loss in KD for recommender systems. Firstly, we extend the connection between CE loss and NDCG to full-item KD, where CE loss is computed using all items. We theoretically prove that minimizing CE loss maximizes the lower bound of NDCG with the relevance scores proportional to the teacher's predicted scores. This suggests a strong motivation for using CE loss in KD.

However, full-item KD is not practical due to the extremely large number of items. In real-world scenarios, CE loss could only be computed on a subset of items (i.e., the partial-item KD), such as the teacher's top items in vanilla CE. For this case, we define partial NDCG, which only considers rankings within a subset of items. Then, we prove that CE loss bounds partial NDCG. However, it holds only if the item subset satisfies our assumption of closure (Assumption 4.3). It requires that all items that the student ranks higher than any item in the subset are also in the subset. This assumption emphasizes the effect of the student's top items. Recall that our goal is to distill rankings among the teacher's top items (Reddi et al., 2021). Unfortunately, we observe that the top items given by the teacher are usually ranked low by the student. This makes it difficult for the teacher's top items to satisfy our assumption of closure. Thus, vanilla CE cannot bound partial NDCG and performs poorly.

To fully unleash the potential of CE loss by re-establishing its connection with NDCG, we propose Rejuvenated Cross-Entropy for Knowledge Distillation (RCE-KD), which consists of four key points: 1) It divides the teacher's top items into two subsets: the subset that consists of items also ranked high by the student and the one that consists of the rest of the items. 2) For the first subset, we distill rankings among these items by using CE loss directly on the student's top items. 3) For the second subset, we design a sampling strategy to sample from the student's top items and compute CE loss on a new item set that approximately satisfies Assumption 4.3. 4) The fusion weights of the losses on these two subsets are adaptively updated based on their size. With the above improvements, we can nearly completely unleash the potential of CE loss while ensuring high training efficiency.

To sum up, the key contributions of our work are as follows:

• We theoretically extend the connection between CE and NDCG to the field of KD for recommender systems in real scenarios, where KD is performed on an item subset. Specifically, we first define partial NDCG, which measures ranking ability on a subset of items. Then, we prove that minimizing CE loss on a given item subset maximizes the partial NDCG on it. We also give the critical assumption made on the item subset for the conclusion to hold.

• Based on the analysis, we propose RCE-KD to unleash the potential of CE loss fully while ensuring high training efficiency. It splits the top items of teachers and calculates the loss separately. A dynamic weighting method is devised to adaptively fuse the losses on all subsets.

• Extensive experiments are conducted on three public datasets and both homogeneous and heterogeneous KD settings to demonstrate the superiority of the proposed approach.

## 2 RELATED WORK

### 2.1 KNOWLEDGE DISTILLATION FOR RECOMMENDER SYSTEMS

Existing KD methods fall into three categories: response-based, feature-based, and relation-based.

**Response-based** methods focus on teachers' predictions. CD (Lee et al., 2019) samples unobserved items from a distribution associated with their rankings predicted by students, and distills with a point-wise loss. RankDistil (Reddi et al., 2021) enables students to mimic teachers by sampling high-ranking items predicted by teachers and calculating multiple forms of loss functions on them. RRD (Kang et al., 2020) adopts a list-wise loss to maximize the likelihood of the teacher's recommendation list. Note that RRD could be regarded as the extension of ListMLE loss (Xia et al., 2008) to the top-K setting (Xia et al., 2009). Based on RRD, DCD (Lee & Kim, 2021) uses the discrepancy between the teacher and student model predictions to decide which knowledge to distill. HetComp (Kang et al., 2023) transfers the ensemble knowledge of heterogeneous teachers by constructing easy-to-hard knowledge sequences from the teachers' trajectories.

**Feature-based** methods focus on the intermediate representations of the teacher. FreqD (Zhu & Zhang, 2024) defines knowledge as different frequency components of the features and proposes emphasizing important knowledge by graph filtering. PCKD (Zhu & Zhang, 2025) observes that projectors in feature-based KD interrupt user preference contained in the features and designs two regularization terms to restrict the projectors.

**Relation-based** methods focus on the relationships between different items. HTD (Kang et al., 2021) distills the sample relation hierarchically to alleviate the capacity gap between the student and teacher.

Our work compensates for the lack of theoretical analysis of CE loss in response-based methods. Based on theoretical analysis, we design a split-and-fusion paradigm with a novel sampling strategy and adaptive loss fusion mechanism to enhance vanilla CE loss, thereby unlocking its full potential.

### 2.2 CONNECTION BETWEEN CE LOSS AND NDCG

Recently, many studies (Cao et al., 2007; Ravikumar et al., 2011; Bruch et al., 2019; Wu et al., 2024; Yang et al., 2024; Xu et al., 2024b) on learning-to-rank (LTR) have focused on the impact of different surrogate loss functions on NDCG. Among them, CE loss is of particular interest due to its wide range of applications. As a pioneer, ListNet (Cao et al., 2007) introduces CE loss into LTR by defining the top-one probability. Then, (Bruch et al., 2019) for the first time proves that CE loss is a bound on NDCG when considering binary ground-truth labels. Subsequently, work has been done to improve CE loss based on this conclusion. For example, PSL (Yang et al., 2024) changes the surrogate activations, and SCE (Xu et al., 2024b) increases the weight of negative samples in CE loss to achieve a tighter bound of NDCG. Another work relevant to us is (Wu et al., 2024). It reveals the pros and cons of sampled CE loss for item recommendation and also relates it to NDCG. However, these methods mentioned above hardly address the case of non-binary ground-truth labels. Moreover, they either do not focus on the scenarios that need item sampling or simply use uniform sampling without making any assumptions about the items being sampled. This makes them entirely inapplicable for KD, where we take the teacher's predictions as labels and emphasize the top-ranked items.

## 3 PRELIMINARY

### 3.1 TOP-$N$ RECOMMENDATION

This work focuses on the top-$N$ recommendation with implicit feedback. Let $\mathcal{U}$ and $\mathcal{I}$ denote the user and item sets, respectively. Then, $|\mathcal{U}|$ and $|\mathcal{I}|$ are the number of users and items, respectively. A recommendation model scores the items not interacted with by the user and recommends $N$ items with the largest scores. We use $r_{ui}$ to denote the score of interaction $(u, i)$ predicted by the recommendation model and use $\boldsymbol{r}_u \in \mathbb{R}^{|\mathcal{I}|}$ to denote the predicted scores of all items for user $u$. In this paper, we use superscripts $S$ and $T$ to denote the student and the teacher, respectively. In the following sections, we default our analysis to any $u \in \mathcal{U}$ if not specified.

## 3.2 Cross-Entropy Loss for Knowledge Distillation

Given an item set $\mathcal{J}^u$ for each user $u \in \mathcal{U}$, CE loss in KD for recommender systems is computed as:

$$\mathcal{L}_{CE} = -\frac{1}{|\mathcal{U}|} \sum_{u \in \mathcal{U}} \sum_{i \in \mathcal{J}^u} \sigma(r_{ui}^T, \mathcal{J}^u) \log \sigma(r_{ui}^S, \mathcal{J}^u), \tag{1}$$

where $r_{ui}^T$ and $r_{ui}^S$ denote the scores predicted by the teacher and the student, respectively. $\sigma(r_{ui}^T, \mathcal{J}^u) = \exp(r_{ui}^T) / \sum_{j \in \mathcal{J}^u} \exp(r_{uj}^T)$ denotes the softmax over item set $\mathcal{J}^u$. Similarly $\sigma(r_{ui}^S, \mathcal{J}^u) = \exp(r_{ui}^S) / \sum_{j \in \mathcal{J}^u} \exp(r_{uj}^S)$. Note that for each user $u$, we only have access to a sampled item subset since it is computationally intractable over the entire item set $\mathcal{I}$ (Sun et al., 2024).

## 4 Connection between CE Loss and Ranking Imitation in KD

This section reveals the connection between CE loss and ranking imitation in the field of KD. As a starting point, we extend the connection between CE and NDCG to the **full-item KD**, where the CE loss is computed using all items. Note that although the conclusion is promising, the full-item KD is not practical due to the extremely large number of items. Therefore, we further analyze the connection between CE loss and partial NDCG in **partial-item KD**, where CE loss is computed using only a subset of items. Finally, we demonstrate the challenges when using CE loss as distillation loss by showing the large differences in the student's and teacher's top items.

### 4.1 Analysis in Full-Item KD

This section studies the full-item KD, where CE loss is computed on the entire item set, i.e., $\mathcal{J}^u = \mathcal{I}$. Given a ground-truth relevance scores vector $\boldsymbol{y} \in \mathbb{R}^{|\mathcal{I}|}$ with $y_i$ denoting the score of item $i$, and the predicted permutation $\boldsymbol{\pi}$, NDCG is defined as:

$$\text{NDCG}(\boldsymbol{\pi}, \boldsymbol{y}) = \frac{\text{DCG}(\boldsymbol{\pi}, \boldsymbol{y})}{\text{DCG}(\widetilde{\boldsymbol{\pi}}, \boldsymbol{y})}, \tag{2}$$

where $\widetilde{\boldsymbol{\pi}}$ is the ideal ranked list (where items are sorted according to $\boldsymbol{y}$). DCG is defined as follows:

$$\text{DCG}(\boldsymbol{\pi}, \boldsymbol{y}) = \sum_{i=1}^{|\mathcal{I}|} \frac{2^{y_i} - 1}{\log_2(1 + \pi^{-1}(i))}, \tag{3}$$

where $\pi^{-1}(i)$ is the rank of item $i$.

In the following theorem, we show that minimizing CE loss maximizes the lower bound of NDCG, where the relevance scores of items are proportional to the scores predicted by the teacher.

**Theorem 4.1.** *Suppose that we compute CE loss on the entire item set $\mathcal{I}$ and take the teacher's predicted scores (i.e., $\boldsymbol{r}_u^T$) as the target. In that case, we maximize a lower bound of NDCG, with the teacher's transformed predictive scores $\boldsymbol{y} = \log_2(\sigma(\boldsymbol{r}_u^T) + 1)$ being the relevance scores. Here $\sigma(\cdot)$ denotes the softmax function and $\sigma(\boldsymbol{r}_u^T)_i = \exp(r_{ui}^T) / \sum_{j \in \mathcal{I}} \exp(r_{uj}^T)$.*

The proof is provided in Appendix B.1. Theorem 4.1 demonstrates that when we minimize CE loss over the entire item set, the student can imitate the teacher in terms of NDCG. This theorem gives an intuitive explanation of the rationality of using CE loss as a distillation loss.

### 4.2 Analysis in Partial-Item KD

Although the above conclusion is promising, we can only afford CE loss with a sampled item subset in real-world scenarios. This section shows that CE loss must involve both the teacher's and the student's predicted top items to make the student benefit from the teacher's ranking ability.

Firstly, we define the partial NDCG to describe NDCG in the partial-item KD scenario. It only focuses on the rankings within the item subset.

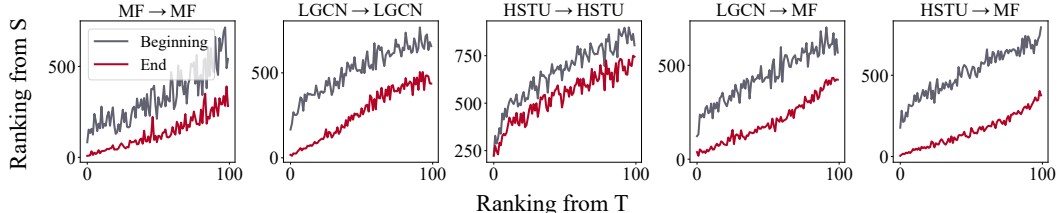

Figure 2: Relationship between rankings given by the teacher (shown in $x$-axis) and the student (shown in $y$-axis). Items are sorted in decreasing order according to the teacher's rankings.

**Definition 4.2** (Partial NDCG). *Given an item set $\mathcal{J}^u$, the partial NDCG on $\mathcal{J}^u$ (denoted as $NDCG_{\mathcal{J}^u}$) is defined as follows:*

$$NDCG_{\mathcal{J}^u}(\boldsymbol{\pi}, \boldsymbol{y}) \triangleq \frac{DCG(\boldsymbol{\pi}, \boldsymbol{y}_{\mathcal{J}^u})}{DCG(\widetilde{\boldsymbol{\pi}}_{\mathcal{J}^u}, \boldsymbol{y}_{\mathcal{J}^u})}, \tag{4}$$

*where*

$$(\boldsymbol{y}_{\mathcal{J}^u})_i = \begin{cases} y_i & \text{if } i \in \mathcal{J}^u, \\ 0 & \text{otherwise,} \end{cases} \tag{5}$$

*denotes the truncated $\boldsymbol{y}$ that only retains the scores corresponding to the items in $\mathcal{J}^u$, and $\widetilde{\boldsymbol{\pi}}_{\mathcal{J}^u}$ is the corresponding ideal ranked list.*

Then, to draw a promising conclusion analogous to the full-item KD, we must make a not mild but critical assumption about the item subset $\mathcal{J}^u$.

**Assumption 4.3** (Closure of $\mathcal{J}^u$). *For each item $i$ in $\mathcal{J}^u$, we assume that all items that are considered by the student to be ranked higher than $i$ are also in $\mathcal{J}^u$. Formally,*

$$\left( \bigcup_{i \in \mathcal{J}^u} \{j | \pi^{-1}(j) \leq \pi^{-1}(i)\} \right) \subseteq \mathcal{J}^u, \tag{6}$$

*where $\pi^{-1}(i)$ is the rank of item $i$ predicted by the student.*

Finally, we have the following theorem that connects CE loss and partial NDCG.

**Theorem 4.4.** *Given an item set $\mathcal{J}^u \subseteq \mathcal{I}$ that satisfies Assumption 4.3, minimizing CE loss on $\mathcal{J}^u$ maximizes a lower bound of $NDCG_{\mathcal{J}^u}$, where the relevance scores are $\boldsymbol{y}_{\mathcal{J}^u} = \left( \log_2(\sigma(\boldsymbol{r}_u^T) + 1) \right)_{\mathcal{J}^u}$.*

The proof is provided in Appendix B.2. Note that Theorem 4.1 can be regarded as a special case of Theorem 4.4 when $\mathcal{J}^u = \mathcal{I}$. Previous works (Kang et al., 2020; Reddi et al., 2021) find that if the student can learn the rankings of top items given by the teacher, it benefits from the teacher's ranking ability. In other words, they expect to connect their distillation losses with $NDCG_{\mathcal{J}^u}$ where $\mathcal{J}^u$ involves the teacher's top items. Our theorem gives a method with theoretical support for accomplishing that purpose. That is, $\mathcal{J}^u$ must also involve enough top items provided by the student.

### 4.3 CHALLENGE IN PARTIAL-ITEM KD

According to our analysis, $\mathcal{J}^u$ must satisfy Assumption 4.3 for the connection between CE loss and partial NDCG to hold. However, it is difficult to satisfy this assumption if we do not explicitly consider the student's top items. Specifically, in Figure 2, we report the relationship between the student's and the teacher's rankings at the beginning and end of the training. The student is trained with vanilla CE loss, which is computed using the teacher's top items. The dataset is CiteULike. Detailed analysis and results on all datasets are given in Appendix A.1. From the results, we find that:

**Observation 4.5.** *The teacher's top items are very likely to be ranked low by the student, especially at the beginning of the training.*

As a result, if we compute CE loss only on the teacher's or the student's top items, we cannot bound partial NDCG on the teacher's top items. Moreover, if we simply add the student's top items to an item subset that initially contains the teacher's top items to make it satisfy the assumption of closure, it will result in a very large item subset.

## 5 REJUVENATED CROSS-ENTROPY FOR KNOWLEDGE DISTILLATION

### 5.1 OVERVIEW OF RCE-KD

To unleash the potential of CE loss of distilling rankings among the teacher's top items, we propose RCE-KD, a novel approach involving both the teacher's and the student's top items in KD. The key is to split the teacher's top items into two subsets based on whether or not an item is ranked high by the student. Then, we try to make both item subsets satisfy Assumption 4.3 exactly or approximately.

Let $\mathcal{Q}_u^T \triangleq \arg \operatorname{top}K(\boldsymbol{r}_u^T)$ and $\mathcal{Q}_u^S \triangleq \arg \operatorname{top}K(\boldsymbol{r}_u^S)$ denote the sets of top-$K$ items predicted by the teacher and the student, respectively. We aim to transfer the teacher's ranking ability over $\mathcal{Q}_u^T$ (Kang et al., 2020; Lee et al., 2019; Tang & Wang, 2018). In RCE-KD, we propose to separate $\mathcal{Q}_u^T$ into two subsets. The first subset is the intersection between $\mathcal{Q}_u^T$ and $\mathcal{Q}_u^S$. The second subset contains the remaining items in $\mathcal{Q}_u^T$. Formally, $(\mathcal{Q}_u^T)_1 \triangleq \mathcal{Q}_u^T \cap \mathcal{Q}_u^S$ and $(\mathcal{Q}_u^T)_2 \triangleq \mathcal{Q}_u^T \backslash (\mathcal{Q}_u^T)_1$.

### 5.2 LOSS FOR $(\mathcal{Q}_u^T)_1$

For the first subset, we transfer the knowledge within it by computing CE loss on $\mathcal{Q}_u^S$. Formally,

$$\mathcal{L}_1 = -\frac{1}{|\mathcal{U}|} \sum_{u \in \mathcal{U}} \sum_{i \in \mathcal{Q}_u^S} \sigma(r_{ui}^T, \mathcal{Q}_u^S) \log \sigma(r_{ui}^S, \mathcal{Q}_u^S). \tag{7}$$

Note that $\mathcal{Q}_u^S$ satisfies Assumption 4.3. Therefore, $\mathcal{L}_1$ can make the student benefit from the teacher's ranking ability by exactly bounding $\text{NDCG}_{\mathcal{Q}_u^S}$. Since $(\mathcal{Q}_u^T)_1$ is a subset of $\mathcal{Q}_u^S$, it encourages the student to learn the rankings among $(\mathcal{Q}_u^T)_1$.

### 5.3 LOSS FOR $(\mathcal{Q}_u^T)_2$

For the second subset, we propose to approximately maximize $\text{NDCG}_{(\mathcal{Q}_u^T)_2}$ by computing CE loss on the union of $(\mathcal{Q}_u^T)_2$ and a set of randomly sampled items. The probability of each item being sampled is defined as follows: For each item $i$ in $(\mathcal{Q}_u^T)_2$, we raise the scores of all items ranked higher than $i$ in the student's predicted ranking by 1. After iterating the entire $(\mathcal{Q}_u^T)_2$, let $z_j$ denote the score of item $j$. Then, the probability of item $j$ to be sampled is given by $p_j \propto e^{z_j/\tau}, \forall j \in \mathcal{I} \backslash \mathcal{Q}_u^T$, where $\tau$ is a hyperparameter and is fixed to 10 in our experiments.

Note that the sampling strategy is adaptive due to: 1) When the student assigns low rankings to all items in $(\mathcal{Q}_u^T)_2$, we sample nearly uniformly from the entire item set $\mathcal{I}$, allowing us to cover more items in multiple training epochs. 2) In contrast, we sample from highly ranked items when the student can already assign higher rankings to items in $(\mathcal{Q}_u^T)_2$. According to Theorem 4.4, these highly ranked items play a greater role in maximizing the partial NDCG on $(\mathcal{Q}_u^T)_2$ and enable us to efficiently approximate the fulfillment of Assumption 4.3.

Using the above sampling strategy, we sample $L$ items and combine them with $(\mathcal{Q}_u^T)_2$ to form the set $\mathcal{A}^u$ (note that we resample in each epoch). Then, CE loss is computed on $\mathcal{A}^u$ as follows:

$$\mathcal{L}_2 = -\frac{1}{|\mathcal{U}|} \sum_{u \in \mathcal{U}} \sum_{i \in \mathcal{A}^u} \sigma(r_{ui}^T, \mathcal{A}^u) \log \sigma(r_{ui}^S, \mathcal{A}^u). \tag{8}$$

### 5.4 ADAPTIVE LOSS FUSION

Note that $\mathcal{L}_1$ and $\mathcal{L}_2$ play different roles. $\mathcal{L}_1$ focuses on $(\mathcal{Q}_u^T)_1$, consisting of top items considered by both the student and teacher. The goal of these items is to distill their fine-grained rankings, which is done by exactly maximize the partial NDCG. On the contrary, $\mathcal{L}_2$ focuses on items not well-mastered by the student. The goal for these items is to improve their rankings, which is done by making the student imitate the teacher's ranking on $(\mathcal{Q}_u^T)_2$ and randomly sampled items.

To combine the two losses, we propose an adaptive weighting scheme. Specifically, the final loss is

$$\mathcal{L}_{RCE-KD} = (1 - \gamma) \cdot \mathcal{L}_1 + \gamma \cdot \mathcal{L}_2, \tag{9}$$

Table 1: Recommendation performance. The best results are in boldface, and the best baselines are underlined. *Improv.b* denotes the relative improvement of RCE-KD over the best baseline. LGCN stands for LightGCN. A paired t-test is performed over 5 independent runs for evaluating $p$-value ($\leq 0.05$ indicates statistical significance).

| T→S | Method | CiteULike | | | | Gowalla | | | | Yelp | | | |
|---|---|---|---|---|---|---|---|---|---|---|---|---|---|
| | | R@10 | N@10 | R@20 | N@20 | R@10 | N@10 | R@20 | N@20 | R@10 | N@10 | R@20 | N@20 |
| MF→MF | Teacher | 0.0283 | 0.0155 | 0.0442 | 0.0198 | 0.1088 | 0.0907 | 0.1544 | 0.1053 | 0.0394 | 0.0253 | 0.0660 | 0.0339 |
| | Student | 0.0177 | 0.0098 | 0.0284 | 0.0128 | 0.0946 | 0.0820 | 0.1329 | 0.0939 | 0.0348 | 0.0222 | 0.0586 | 0.0299 |
| | CD | 0.0239 | 0.0131 | 0.0347 | 0.0158 | 0.0979 | 0.0855 | 0.1389 | 0.0977 | 0.0370 | 0.0236 | 0.0608 | 0.0310 |
| | RRD | 0.0251 | 0.0135 | 0.0362 | 0.0169 | 0.0977 | 0.0861 | 0.1395 | 0.0987 | 0.0362 | 0.0230 | 0.0626 | 0.0319 |
| | DCD | 0.0254 | 0.0136 | 0.0375 | 0.0173 | 0.1007 | 0.0871 | 0.1413 | 0.0999 | 0.0377 | 0.0240 | 0.0639 | 0.0330 |
| | HetComp | 0.0255 | 0.0135 | 0.0391 | 0.0177 | 0.1028 | 0.0874 | 0.1427 | 0.1010 | 0.0383 | 0.0245 | 0.0644 | 0.0331 |
| | TARec | 0.0254 | 0.0133 | 0.0380 | 0.0171 | 0.1012 | 0.0870 | 0.1416 | 0.0993 | 0.0380 | 0.0241 | 0.0635 | 0.0327 |
| | RCE-KD | **0.0278** | **0.0152** | **0.0431** | **0.0194** | **0.1082** | **0.0905** | **0.1525** | **0.1047** | **0.0400** | **0.0259** | **0.0667** | **0.0345** |
| | *Improv.b* | 9.02% | 11.76% | 10.23% | 9.60% | 5.25% | 3.55% | 6.87% | 3.66% | 4.44% | 5.71% | 3.57% | 4.23% |
| | *p-value* | 1.71e-4 | 1.98e-5 | 5.47e-4 | 7.12e-5 | 4.22e-5 | 2.79e-4 | 8.22e-4 | 3.37e-3 | 6.32e-4 | 1.22e-4 | 1.72e-3 | 3.17e-5 |
| LGCN→LGCN | Teacher | 0.0296 | 0.0160 | 0.0461 | 0.0205 | 0.1236 | 0.1035 | 0.1730 | 0.1190 | 0.0432 | 0.0276 | 0.0716 | 0.0367 |
| | Student | 0.0215 | 0.0113 | 0.0344 | 0.0148 | 0.1098 | 0.0928 | 0.1550 | 0.1069 | 0.0363 | 0.0235 | 0.0621 | 0.0308 |
| | CD | 0.0234 | 0.0125 | 0.0354 | 0.0161 | 0.1132 | 0.0951 | 0.1592 | 0.1100 | 0.0385 | 0.0247 | 0.0669 | 0.0342 |
| | RRD | 0.0247 | 0.0125 | 0.0359 | 0.0158 | 0.1142 | 0.0969 | 0.1627 | 0.1109 | 0.0391 | 0.0245 | 0.0671 | 0.0338 |
| | DCD | 0.0243 | 0.0124 | 0.0360 | 0.0155 | 0.1149 | 0.0971 | 0.1631 | 0.1108 | 0.0403 | 0.0247 | 0.0676 | 0.0340 |
| | HetComp | 0.0248 | 0.0127 | 0.0362 | 0.0160 | 0.1150 | 0.0981 | 0.1636 | 0.1120 | 0.0405 | 0.0256 | 0.0691 | 0.0350 |
| | TARec | 0.0231 | 0.0121 | 0.0350 | 0.0157 | 0.1141 | 0.0971 | 0.1630 | 0.1112 | 0.0409 | 0.0258 | 0.0694 | 0.0352 |
| | RCE-KD | **0.0262** | **0.0133** | **0.0377** | **0.0171** | **0.1196** | **0.1011** | **0.1681** | **0.1163** | **0.0431** | **0.0277** | **0.0716** | **0.0369** |
| | *Improv.b* | 5.65% | 4.73% | 4.12% | 6.21% | 4.00% | 3.06% | 2.75% | 3.84% | 5.38% | 7.36% | 3.17% | 4.83% |
| | *p-value* | 4.72e-4 | 7.33e-4 | 9.87e-4 | 3.73e-3 | 1.31e-3 | 4.52e-4 | 2.01e-3 | 7.38e-4 | 4.52e-3 | 3.37e-3 | 8.66e-4 | 7.92e-4 |
| HSTU→HSTU | Teacher | 0.0463 | 0.0291 | 0.0613 | 0.0333 | 0.1124 | 0.0901 | 0.1625 | 0.1063 | 0.0482 | 0.0314 | 0.0800 | 0.0417 |
| | Student | 0.0262 | 0.0159 | 0.0371 | 0.0189 | 0.0974 | 0.0781 | 0.1416 | 0.0923 | 0.0391 | 0.0254 | 0.0664 | 0.0342 |
| | CD | 0.0428 | 0.0264 | 0.0565 | 0.0285 | 0.1029 | 0.0817 | 0.1527 | 0.0997 | 0.0415 | 0.0271 | 0.0701 | 0.0365 |
| | RRD | 0.0433 | 0.0263 | 0.0562 | 0.0297 | 0.1048 | 0.0835 | 0.1538 | 0.1002 | 0.0433 | 0.0274 | 0.0712 | 0.0373 |
| | DCD | 0.0461 | 0.0292 | 0.0599 | 0.0319 | 0.1060 | 0.0857 | 0.1541 | 0.1021 | 0.0443 | 0.0290 | 0.0728 | 0.0382 |
| | HetComp | 0.0470 | 0.0299 | 0.0609 | 0.0331 | 0.1049 | 0.0840 | 0.1532 | 0.1018 | 0.0440 | 0.0285 | 0.0719 | 0.0377 |
| | TARec | 0.0447 | 0.0270 | 0.0571 | 0.0299 | 0.1053 | 0.0842 | 0.1537 | 0.1005 | 0.0437 | 0.0280 | 0.0713 | 0.0369 |
| | RCE-KD | **0.0524** | **0.0325** | **0.0670** | **0.0366** | **0.1106** | **0.0902** | **0.1594** | **0.1058** | **0.0459** | **0.0305** | **0.0754** | **0.0400** |
| | *Improv.b* | 11.49% | 8.70% | 10.02% | 10.57% | 4.34% | 5.25% | 3.44% | 3.62% | 3.61% | 5.17% | 3.57% | 4.71% |
| | *p-value* | 1.73e-4 | 4.22e-4 | 8.92e-5 | 3.77e-4 | 3.52e-4 | 9.92e-4 | 4.57e-3 | 8.20e-3 | 7.32e-4 | 3.71e-5 | 2.23e-3 | 2.38e-4 |
| LGCN→MF | Teacher | 0.0296 | 0.0160 | 0.0461 | 0.0205 | 0.1236 | 0.1035 | 0.1730 | 0.1190 | 0.0432 | 0.0276 | 0.0716 | 0.0367 |
| | Student | 0.0177 | 0.0098 | 0.0284 | 0.0128 | 0.0946 | 0.0820 | 0.1329 | 0.0939 | 0.0348 | 0.0222 | 0.0586 | 0.0299 |
| | CD | 0.0240 | 0.0133 | 0.0365 | 0.0170 | 0.1097 | 0.0917 | 0.1549 | 0.1072 | 0.0368 | 0.0247 | 0.0610 | 0.0342 |
| | RRD | 0.0247 | 0.0137 | 0.0367 | 0.0169 | 0.1098 | 0.0932 | 0.1577 | 0.1070 | 0.0377 | 0.0249 | 0.0622 | 0.0340 |
| | DCD | 0.0260 | 0.0139 | 0.0387 | 0.0177 | 0.1123 | 0.0966 | 0.1600 | 0.1098 | 0.0392 | 0.0258 | 0.0641 | 0.0347 |
| | HetComp | 0.0263 | 0.0142 | 0.0402 | 0.0180 | 0.1110 | 0.0943 | 0.1604 | 0.1103 | 0.0399 | 0.0260 | 0.0657 | 0.0344 |
| | TARec | 0.0254 | 0.0137 | 0.370 | 0.0173 | 0.1112 | 0.0950 | 0.1603 | 0.1100 | 0.0406 | 0.0263 | 0.0663 | 0.0350 |
| | RCE-KD | **0.0285** | **0.0156** | **0.0437** | **0.0197** | **0.1200** | **0.1013** | **0.1677** | **0.1163** | **0.0419** | **0.0271** | **0.0692** | **0.0360** |
| | *Improv.b* | 8.37% | 9.86% | 8.71% | 9.44% | 6.86% | 4.87% | 4.55% | 5.44% | 2.95% | 3.04% | 4.37% | 2.86% |
| | *p-value* | 1.77e-5 | 1.92e-4 | 4.29e-4 | 6.99e-5 | 4.33e-5 | 3.52e-4 | 1.21e-3 | 3.23e-4 | 7.71e-4 | 6.82e-3 | 9.95e-4 | 1.03e-3 |
| HSTU→MF | Teacher | 0.0463 | 0.0291 | 0.0613 | 0.0333 | 0.1124 | 0.0901 | 0.1625 | 0.1063 | 0.0482 | 0.0314 | 0.0800 | 0.0417 |
| | Student | 0.0177 | 0.0098 | 0.0284 | 0.0128 | 0.0946 | 0.0820 | 0.1329 | 0.0939 | 0.0348 | 0.0222 | 0.0586 | 0.0299 |
| | CD | 0.0361 | 0.0209 | 0.0502 | 0.0251 | 0.1047 | 0.0834 | 0.1520 | 0.1021 | 0.0433 | 0.0276 | 0.0743 | 0.0370 |
| | RRD | 0.0379 | 0.0224 | 0.0520 | 0.0270 | 0.1054 | 0.0831 | 0.1511 | 0.1018 | 0.0430 | 0.0271 | 0.0734 | 0.0359 |
| | DCD | 0.0411 | 0.0253 | 0.0533 | 0.0295 | 0.1078 | 0.0854 | 0.1552 | 0.1029 | 0.0449 | 0.0297 | 0.0759 | 0.0392 |
| | HetComp | 0.0401 | 0.0239 | 0.0524 | 0.0278 | 0.1066 | 0.0840 | 0.1531 | 0.1031 | 0.0453 | 0.0290 | 0.0765 | 0.0382 |
| | TARec | 0.0416 | 0.0258 | 0.0538 | 0.0300 | 0.1059 | 0.0838 | 0.1527 | 0.0123 | 0.0437 | 0.0293 | 0.0762 | 0.0378 |
| | RCE-KD | **0.0457** | **0.0285** | **0.0595** | **0.0324** | **0.1128** | **0.0905** | **0.1624** | **0.1065** | **0.0485** | **0.0316** | **0.0805** | **0.0419** |
| | *Improv.b* | 9.86% | 10.47% | 10.59% | 8.00% | 4.64% | 5.97% | 4.64% | 3.30% | 7.06% | 6.40% | 5.23% | 6.89% |
| | *p-value* | 6.73e-5 | 5.66e-4 | 1.93e-4 | 7.11e-5 | 1.37e-3 | 2.52e-4 | 3.31e-4 | 2.83e-3 | 1.07e-4 | 3.88e-4 | 1.17e-3 | 4.92e-4 |

where $\gamma$ is updated at the beginning of each epoch by the following equation:

$$\gamma = \exp\left(-\beta \cdot \frac{|(\mathcal{Q}_u^T)_1|}{|\mathcal{Q}_u^T|}\right), \tag{10}$$

where $|\cdot|$ denotes the cardinality of the set and $\beta$ is a hyperparameter. When $|(\mathcal{Q}_u^T)_1|$ is small, we make the student overlap more with the teacher's top items by increasing the weight of $\mathcal{L}_2$. Otherwise, we assign a large weight to $\mathcal{L}_1$ because it is more useful when we want to distill fine-grained rankings.

In Appendix C.5, we provide extensive, multi-level experiments that thoroughly validate the superiority of our adaptive $\gamma$-scheduling strategy.

Finally, the total loss for training the student is given by

$$\mathcal{L} = \mathcal{L}_{Base} + \lambda \cdot \mathcal{L}_{RCE-KD}, \tag{11}$$

where $\mathcal{L}_{Base}$ is the loss of the base recommendation model, such as BPR loss. $\lambda$ is a hyperparameter.

## 6 EXPERIMENTS

Section 6.1 first introduces the experimental settings. The **implementation details** are shown in Appendix C.1. Then, the overall performance comparison is shown in Section 6.2. Consequently,

we investigate the training efficiency of all compared KD methods in Section 6.3. The ablation study is conducted in Section 6.4. To verify our sampling strategy's **efficiency for approximating Assumption 4.3**, we conduct experiments in Appendix C.2. We present **hyperparameter analysis** in Appendix C.4. Then, we provide multi-level experiments in Appendix C.5 to validate the superiority of our adaptive $\gamma$-scheduling strategy. Moreover, in Appendix D.1 and Appendix D.2, we demonstrate the effectiveness of **applying our method to sequential recommendation and multi-modal recommendation**, respectively, to showcase its generalization capability for recommendation tasks. Finally, we also **visualized the evolution of NDCG during training** in Appendix A.2 to demonstrate that RCE-KD successfully bounds NDCG as theoretically expected.

## 6.1 EXPERIMENTAL SETTINGS

**Datasets.** We conduct experiments on three public datasets, including **CiteULike** (Wang et al., 2013; Kang et al., 2022; 2021), **Gowalla** (Cho et al., 2011; Tang & Wang, 2018; Lee et al., 2019), and **Yelp2018** (Lee et al., 2019; Kweon et al., 2021). Detailed statistics and methods of constructing training and test sets are given in Appendix C.1.

**Evaluation Protocols.** Per the custom, we adopt the full-ranking evaluation to achieve an unbiased evaluation. We employ Recall (Recall@$N$) and normalized discounted cumulative gain (NDCG@$N$) and report the results for $N \in \{10, 20\}$. We conduct five independent runs for each configuration and report the average results.

**Baselines.** We compare our method with five response-based KD methods: CD (Lee et al., 2019), RRD (Kang et al., 2020), DCD (Lee & Kim, 2021), HetComp (Kang et al., 2023), and TARec (Zhuang et al., 2025). The introduction of these methods is in Appendix C.1.

**Backbones.** We refer to previous works (Chen et al., 2023; Kang et al., 2020; 2021), and use MF (Rendle et al., 2012) and LightGCN (He et al., 2020). We also add HSTU (Zhai et al., 2024) as a new backbone, which is a popular generative recommendation model.

**Teacher/Student.** For each backbone, we create two instances, one large and one small. We use the large instance as the teacher and the small one as the student. Details are provided in Appendix C.1.

## 6.2 PERFORMANCE COMPARISON

The performance of all methods is provided in Table 1. From the results, we observe that:
• Different KD methods perform differently. We find that CD performs poorly compared to other methods. We attribute this to CD using a pair-wise loss to align teachers' and students' predictions. In training recommendation models, pair-wise loss is usually less effective than list-wise losses, such as RRD loss and CE loss.
• Our method significantly outperforms all other methods in all cases, suggesting it effectively aligns the teacher's and student's predictions and utilizes the teacher's predictions to enhance the student. This also demonstrates that utilizing CE loss for KD and using teacher and student predictions to collaboratively decide on sampling strategies are effective.
• In all scenarios, students can perform similarly to teachers. This suggests that with the proper knowledge distillation approach, we can significantly reduce the model size and improve the model's inference efficiency with little to no degradation of the model's recommendation accuracy.

Table 2: The comparison of the training time (seconds) per epoch.

| Method | CiteULike | | | Gowalla | | | Yelp | | |
|---|---|---|---|---|---|---|---|---|---|
| | MF | LightGCN | HSTU | MF | LightGCN | HSTU | MF | LightGCN | HSTU |
| Student | 4.25 | 5.47 | 8.12 | 27.33 | 50.11 | 58.72 | 26.93 | 36.53 | 49.29 |
| CD | 14.37 | 20.80 | 29.03 | 82.77 | 137.63 | 201.70 | 77.72 | 126.38 | 210.69 |
| RRD | 19.67 | 23.81 | 39.09 | 132.37 | 167.90 | 231.12 | 119.49 | 152.27 | 271.86 |
| DCD | 21.37 | 26.82 | 38.63 | 145.87 | 158.60 | 241.35 | 108.37 | 162.92 | 292.00 |
| HetComp | 16.32 | 24.87 | 40.03 | 137.62 | 144.53 | 239.07 | 121.36 | 156.04 | 281.91 |
| RCE-KD | 15.23 | 21.79 | 33.62 | 99.30 | 141.72 | 221.65 | 81.17 | 129.74 | 233.38 |

Table 3: The comparison of GPU Memory (GB) required by our method and comparison methods.

| Method | CiteULike | | | Gowalla | | | Yelp | | |
|---|---|---|---|---|---|---|---|---|---|
| | MF | LightGCN | HSTU | MF | LightGCN | HSTU | MF | LightGCN | HSTU |
| Student | 0.39 | 0.60 | 3.11 | 0.45 | 1.08 | 1.10 | 0.45 | 0.81 | 1.36 |
| CD | 1.07 | 2.52 | 6.27 | 6.27 | 8.81 | 19.37 | 4.99 | 7.02 | 17.62 |
| RRD | 0.92 | 2.42 | 6.81 | 5.23 | 8.64 | 19.93 | 4.85 | 6.30 | 19.01 |
| DCD | 1.41 | 2.93 | 7.22 | 7.89 | 9.37 | 21.52 | 6.03 | 7.22 | 20.89 |
| HetComp | 1.09 | 2.69 | 6.97 | 6.78 | 9.02 | 19.99 | 5.87 | 7.01 | 19.97 |
| RCE-KD | 1.05 | 2.47 | 6.52 | 6.37 | 8.90 | 19.98 | 5.41 | 6.90 | 18.74 |

## 6.3 TRAINING EFFICIENCY

In this section, we report the training efficiency of our method and comparison methods. Since TARec involves a two-stage training process, we did not include it in the comparison. All results are obtained by testing with PyTorch on a GeForce RTX 3090 GPU.

In RCE-KD, we only need to add the cost of time and space required for random sampling on top of CE loss. Therefore, it has very high training efficiency. To empirically validate the training efficiency of our method, we report the training time and storage cost of our method and comparison methods. The results are presented in Table 2 and Table 3. The method *Student* denotes that we train the student model without KD. Note that since we save the teacher's predictions before KD and simply load the predictions without rerunning the teacher during KD, the architecture of the teachers does not affect the training inference. Therefore, we only report the results with different students.

From the results, we find that:

- All KD methods inevitably increase training costs. In most cases, all KD methods have similar training costs. We believe this is attributed to the fact that they all follow a similar pattern of sampling a subset of items before computing the loss functions.

- Among all baseline methods, we find that CD and RRD have smaller training costs than others. We believe this is because CD and RRD are simpler and require fewer intermediate computational processes. However, they do not perform as well as the more complex methods. This forces baselines to face a trade-off between training cost and recommendation accuracy.

- Our method has similar training efficiency as CD and RRD. This can be attributed to the simplicity of our method. Moreover, we empirically find that the number of items that need to be sampled by our method is often smaller than that of other methods, significantly reducing the cost required in the sampling phase. Together, these two make our method highly efficient in training.

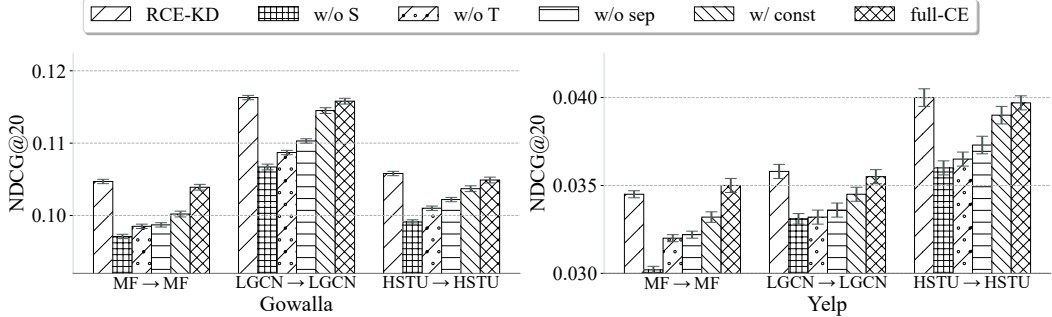

Figure 3: Ablation study on Gowalla and Yelp, including the results in three homogeneous Teacher → Student settings.

## 6.4 ABLATION STUDY

RCE-KD consists of three key components: 1) It divides the teacher's top items into two subsets; 2) It computes CE loss on items selected from both the teacher's and the student's top items; 3) An adaptive mechanism is proposed to combine the losses on these subsets. To validate the effectiveness of these key components, we design four variants: 1) **RCE-KD w/o sep** does not compute losses for

$(\mathcal{Q}_u^T)_1$ and $(\mathcal{Q}_u^T)_1$, separately. It only computes CE loss on $\mathcal{A}^u \cup \mathcal{Q}_u^S$; 2) **RCE-KD w/o S** only aligns the predictions on the teacher's predicted top items, i.e., $\mathcal{Q}_u^T$; 3) Similarly, **RCE-KD w/o T** only aligns the predictions on the student's predicted top items, i.e., $\mathcal{Q}_u^S$; 4) **RCE-KD w/ const** replaces the adaptive weight derived from Eq.(10) with a constant hyperparameter $\gamma$. We also compare with CE loss computed on the full item set, denoted as **full-CE**, to validate the effectiveness of our method.

Figure 3 shows the results of these four variants on Gowalla and Yelp, and three Teacher/Student settings. The results of the remaining settings are provided in Appendix C.3. We find that all variants are inferior to the original RCE-KD, which demonstrates the effectiveness of all key components. Moreover, **RCE-KD w/o S** usually performs worse than **RCE-KD w/o T**. We believe that the reason is that the top items given by the student can exactly satisfy Assumption 4.3, while the top items given by the teacher do not. The superiority of **RCE-KD w/ const** over **RCE-KD w/o T** demonstrates the necessity of involving top items from both the student and the teacher. The superiority of RCE-KD over **RCE-KD w/ const** and **RCE-KD w/o sep** validates the effectiveness of our adaptive weighting scheme and the necessity of splitting out the two subsets and treating them separately. Finally, our method performs even slightly above **full-CE** in most cases, due to a tighter bound on NDCG than full-item CE (Xu et al., 2024a).

## 7 CONCLUSION

This paper analyzes CE loss in the real KD scenario for recommender systems, where loss is computed using a subset of items. We prove that CE loss bounds NDCG. It makes CE loss suitable for recommender systems, where rankings are essential. We also theoretically provide a critical assumption about the item subset, on which CE loss is computed, for the conclusion to hold. Based on the above analysis, we propose RCE-KD to fully unleash the potential of CE loss by approximately satisfying the assumption through teacher-student collaboration. Extensive experiments on both homogeneous and heterogeneous settings demonstrate the effectiveness of our method.

## ACKNOWLEDGMENTS

This work was supported in part by National Natural Science Foundation of China (No. 62572198 and 92270119).

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

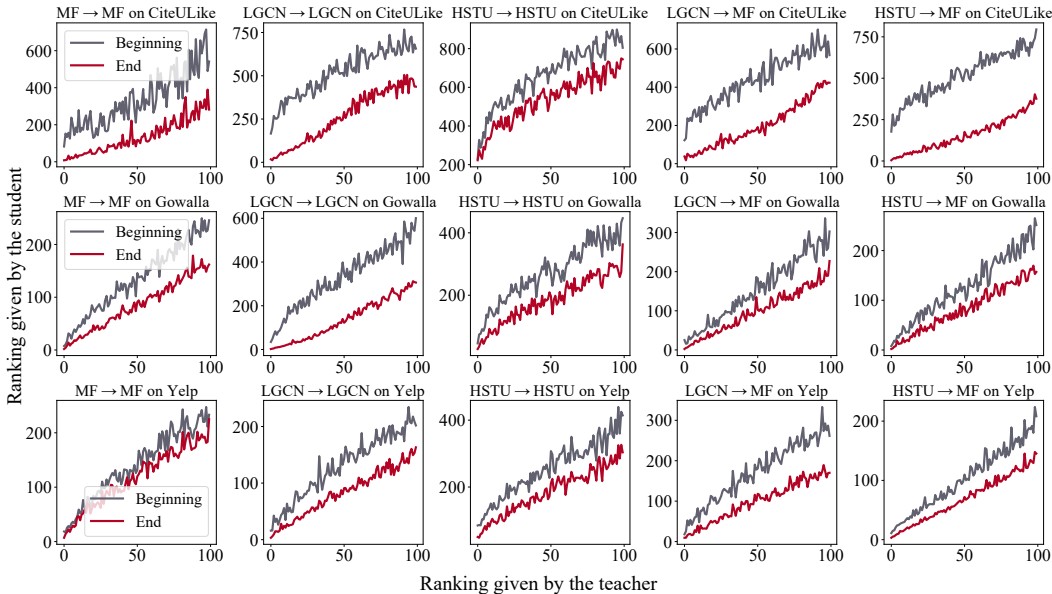

Figure 4: Relationship between rankings given by the teacher (shown in $x$-axis) and the student (shown in $y$-axis) on all datasets. Items are sorted in decreasing order according to the teacher's rankings.

## A  VISUALIZATIONS

### A.1  RELATIONSHIP BETWEEN THE RANKINGS GIVEN BY THE STUDENT AND THE TEACHER

To investigate whether we can satisfy Assumption 4.3 without explicitly considering the student's top items, we report the rankings given by the student and the teacher. The student is trained with vanilla CE loss, which is computed using the teacher's top items. The items are sorted in decreasing order according to the rankings given by the teacher. The results on all three datasets are provided in Figure 4. In each subfigure, we give two lines. The grey one represents the results at the beginning of training (after about $0.2\times$ total epoch number of rounds of training). The red one represents the results after training is complete.

The results show similar trends in all cases. Concretely, we observe that: 1) There is a significant positive correlation between the rankings given by the teacher and the student. This suggests that through knowledge distillation, students do learn some of the teacher's ranking results. 2) For top items given by the teacher (ranked higher than 100), students often give lower rankings (lower than 100 and even 200 on CiteULike). 3) The phenomenon is particularly acute at the beginning of training.

### A.2  TIGHTER NDCG BOUND VERIFICATION

To empirically verify that our improvement to the CE loss indeed enables us to bound NDCG more tightly, as predicted by our theory, we conduct a comprehensive analysis of the training dynamics across all three recommendation datasets: CiteULike, Gowalla, and Yelp. Specifically, we monitor the NDCG@10 metric on the training set throughout the training process, comparing our improved CE loss with the original CE loss under various teacher-student architecture combinations.

Figure 5 presents the training curves of NDCG@10 for both methods across five distinct knowledge distillation scenarios: MF→MF, LGCN→LGCN, HSTU→HSTU, LGCN→MF, and HSTU→MF. The experimental results consistently demonstrate three key advantages of our RCE-KD:

**Higher NDCG values:** Across all datasets and teacher-student combinations, RCE-KD consistently achieves higher NDCG@10 values throughout the training process compared to Vanilla CE. This indicates that our approach consistently and stably improves NDCG throughout the entire training

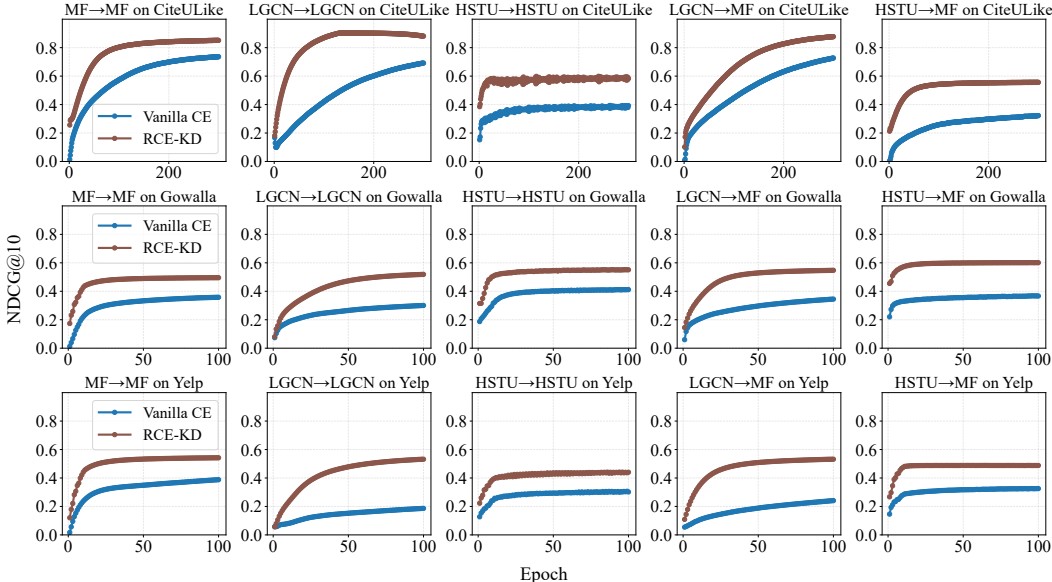

Figure 5: Training curves of NDCG@10 on the training set for RCE-KD and Vanilla CE across three datasets (CiteULike, Gowalla, Yelp) and five teacher-student architecture combinations.

process. Moreover, the higher NDCG achieved by our method after training convergence demonstrates its superior ability to optimize NDCG.

**Faster growth rate:** The training curves reveal that RCE-KD exhibits a steeper learning curve, indicating a more rapid improvement in NDCG during the early training epochs. This accelerated learning can be attributed to the more informative gradient signals provided by RCE-KD. This observation directly supports our theoretical analysis that RCE-KD provides a tighter bound on the ranking metric, as reflected by the superior optimization of NDCG during training.

**Faster convergence:** In addition to achieving higher peak performance, RCE-KD demonstrates faster convergence to its optimal NDCG value in most cases. The training curves show that RCE-KD reaches its plateau earlier than Vanilla CE, suggesting that the method not only achieves better final performance but also requires fewer training epochs to reach convergence.

Based on the above observations, we can confidently assert that our method indeed achieves tighter bounds on NDCG, as predicted by our theoretical analysis. Consequently, it enables students to learn the ranking capabilities of teachers more effectively, making it more suitable for knowledge distillation in recommender systems.

# B PROOFS

## B.1 PROOF OF THEOREM 4.1

*Proof.* By inserting $\boldsymbol{y} = \log_2(\sigma(\boldsymbol{r}_u^T) + 1)$ into the definition of DCG in Eq.(3), we have

$$\mathrm{DCG}(\boldsymbol{\pi}, \boldsymbol{y}) = \sum_{i \in \mathcal{I}} \frac{\sigma(\boldsymbol{r}_u^T)_i}{\log_2(1 + \pi^{-1}(i))}. \tag{12}$$

Then, similar to the proof for Theorem 3 in (Bruch et al., 2019), we have

$$\text{DCG}(\boldsymbol{\pi}, \boldsymbol{y}) = \sum_{i \in \mathcal{I}} \frac{\sigma(\boldsymbol{r}_u^T)_i}{\log_2(1 + \pi^{-1}(i))} \tag{13}$$

$$\geq \sum_{i \in \mathcal{I}} \frac{\sigma(\boldsymbol{r}_u^T)_i}{\pi^{-1}(i)} \tag{14}$$

$$\geq \sum_{i \in \mathcal{I}} \sigma(\boldsymbol{r}_u^T)_i \cdot \frac{\exp(r_{ui}^S)}{\sum_{k \in \mathcal{I}} \exp(r_{uk}^S)}, \tag{15}$$

where $\boldsymbol{r}_u^S$ is the student's predictive score vector that derives the permutation $\boldsymbol{\pi}$.

For the ideal DCG, we have

$$\text{DCG}(\widetilde{\boldsymbol{\pi}}, \boldsymbol{y}) = \sum_{i \in \mathcal{I}} \frac{\sigma(\boldsymbol{r}_u^T)_i}{\log_2(1 + \widetilde{\pi}^{-1}(i))} \tag{16}$$

$$\leq \sum_{i \in \mathcal{I}} \sigma(\boldsymbol{r}_u^T)_i \tag{17}$$

$$= 1 \tag{18}$$

Finally,

$$\log \text{NDCG}(\boldsymbol{\pi}, \boldsymbol{y}) = \log \left( \frac{\text{DCG}(\boldsymbol{\pi}, \boldsymbol{y})}{\text{DCG}(\widetilde{\boldsymbol{\pi}}, \boldsymbol{y})} \right) \tag{19}$$

$$\geq \log \left( \sum_{i \in \mathcal{I}} \sigma(\boldsymbol{r}_u^T)_i \cdot \frac{\exp(r_{ui}^S)}{\sum_{k \in \mathcal{I}} \exp(r_{uk}^S)} \right) \tag{20}$$

$$\geq \sum_{i \in \mathcal{I}} \sigma(\boldsymbol{r}_u^T)_i \log \left( \frac{\exp(r_{ui}^S)}{\sum_{k \in \mathcal{I}} \exp(r_{uk}^S)} \right), \tag{21}$$

where the final inequality holds because of Jensen's inequality. We complete the proof by noting that the right-hand side of the final inequality is the negative of CE loss. □

### B.2 PROOF OF THEOREM 4.4

*Proof.*

$$\text{NDCG}_{\mathcal{J}^u}(\boldsymbol{\pi}, \boldsymbol{y}) = \frac{\text{DCG}(\boldsymbol{\pi}, \boldsymbol{y}_{\mathcal{J}^u})}{\text{DCG}(\widetilde{\boldsymbol{\pi}}_{\mathcal{J}^u}, \boldsymbol{y}_{\mathcal{J}^u})} \tag{22}$$

$$\geq \text{DCG}(\boldsymbol{\pi}, \boldsymbol{y}_{\mathcal{J}^u}) \qquad \text{(Due to } \text{DCG}(\widetilde{\boldsymbol{\pi}}_{\mathcal{J}^u}, \boldsymbol{y}_{\mathcal{J}^u}) \leq 1.)$$

$$= \sum_{i \in \mathcal{J}^u} \frac{\sigma(\boldsymbol{r}_u^T)_i}{\log_2(1 + \pi^{-1}(i))} \tag{23}$$

$$\geq \sum_{i \in \mathcal{J}^u} \sigma(\boldsymbol{r}_u^T)_i \cdot \frac{1}{\pi^{-1}(i)} \tag{24}$$

$$\geq \sum_{i \in \mathcal{J}^u} \sigma(\boldsymbol{r}_u^T)_i \cdot \frac{1}{\sum_{\pi^{-1}(j) \leq \pi^{-1}(i)} \exp(r_{uj}^S - r_{ui}^S)} \tag{25}$$

$$= \sum_{i \in \mathcal{J}^u} \sigma(\boldsymbol{r}_u^T)_i \cdot \frac{\exp(r_{ui}^S)}{\sum_{\pi^{-1}(j) \leq \pi^{-1}(i)} \exp(r_{uj}^S)} \tag{26}$$

$$\geq \sum_{i \in \mathcal{J}^u} \sigma(\boldsymbol{r}_u^T)_i \cdot \frac{\exp(r_{ui}^S)}{\sum_{j \in \mathcal{J}^u} \exp(r_{uj}^S)}. \tag{27}$$

Table 4: Statistics of the preprocessed datasets.

| Dataset | #Users | #Items | #Interactions | #Sparsity |
|---|---|---|---|---|
| CiteULike | 5,219 | 25,181 | 125,580 | 99.89% |
| Gowalla | 29,858 | 40,981 | 1,027,370 | 99.92% |
| Yelp2018 | 41,801 | 26,512 | 1,022,604 | 99.91% |

Table 5: Dimensions of teachers and students for MF and LightGCN.

| Model | CiteULike | | Gowalla | | Yelp | |
|---|---|---|---|---|---|---|
| | MF | LightGCN | MF | LightGCN | MF | LightGCN |
| Teacher | 400 | 2000 | 300 | 2000 | 300 | 1000 |
| Student | 20 | 20 | 20 | 20 | 20 | 20 |

Therefore,

$$\log \text{NDCG}_{\mathcal{J}^u}(\boldsymbol{\pi}, \boldsymbol{y}) \geq \log \sum_{i \in \mathcal{J}^u} \sigma(\boldsymbol{r}_u^T)_i \cdot \frac{\exp(r_{ui}^S)}{\sum_{j \in \mathcal{J}^u} \exp(r_{uj}^S)} \tag{28}$$

$$= \log \sum_{i \in \mathcal{J}^u} \frac{\exp(r_{ui}^T)}{\sum_{j \in \mathcal{J}^u} \exp(r_{uj}^T)} \cdot \frac{\exp(r_{ui}^S)}{\sum_{j \in \mathcal{J}^u} \exp(r_{uj}^S)} + \log \sum_{j \in \mathcal{J}^u} \sigma(\boldsymbol{r}_u^T)_j \tag{29}$$

$$\geq \sum_{i \in \mathcal{J}^u} \frac{\exp(r_{ui}^T)}{\sum_{j \in \mathcal{J}^u} \exp(r_{uj}^T)} \log \frac{\exp(r_{ui}^S)}{\sum_{j \in \mathcal{J}^u} \exp(r_{uj}^S)} + \log \sum_{j \in \mathcal{J}^u} \sigma(\boldsymbol{r}_u^T)_j \tag{30}$$

$$= \sum_{i \in \mathcal{J}^u} \frac{\exp(r_{ui}^T)}{\sum_{j \in \mathcal{J}^u} \exp(r_{uj}^T)} \log \frac{\exp(r_{ui}^S)}{\sum_{j \in \mathcal{J}^u} \exp(r_{uj}^S)} + \log C_{\mathcal{J}^u}, \tag{31}$$

where $C_{\mathcal{J}^u} \triangleq \sum_{j \in \mathcal{J}^u} \sigma(\boldsymbol{r}_u^T)_j$ is a constant, given $\mathcal{J}^u$.

Note that by minimizing CE loss on $\mathcal{J}^u$, which is defined as follows:

$$-\sum_{i \in \mathcal{J}^u} \frac{\exp(r_{ui}^T)}{\sum_{j \in \mathcal{J}^u} \exp(r_{uj}^T)} \log \frac{\exp(r_{ui}^S)}{\sum_{j \in \mathcal{J}^u} \exp(r_{uj}^S)}, \tag{32}$$

we also maximize

$$\sum_{i \in \mathcal{J}^u} \frac{\exp(r_{ui}^T)}{\sum_{j \in \mathcal{J}^u} \exp(r_{uj}^T)} \log \frac{\exp(r_{ui}^S)}{\sum_{j \in \mathcal{J}^u} \exp(r_{uj}^S)} + \log C_{\mathcal{J}^u}, \tag{33}$$

because $C_{\mathcal{J}^u}$ is a constant when $\mathcal{J}^u$ is fixed. $\qquad\square$

## C MORE EXPERIMENTAL RESULTS

### C.1 EXPERIMENTAL SETTINGS

**Datasets.** We conduct experiments on three public datasets, including **CiteULike**[1] (Wang et al., 2013; Kang et al., 2022; 2021), **Gowalla**[2] (Cho et al., 2011; Tang & Wang, 2018; Lee et al., 2019), and **Yelp2018**[3] (Lee et al., 2019; Kweon et al., 2021).

Following the previous method (Xu et al., 2023), we filter out users and items with less than 10 interactions and then split the rest chronologically into training, validation, and test sets in an 8:1:1 ratio. The statistics of the preprocessed datasets are summarized in Table 4.

---

[1]https://github.com/changun/CollMetric/tree/master/citeulike-t
[2]http://dawenl.github.io/data/gowallapro.zip
[3]https://github.com/hexiangnan/sigir16-eals

Table 6: The Number of transformer blocks (#Block) and number of heads (#Head) for HSTU.

| Model | CiteULike | | Gowalla | | Yelp | |
|---|---|---|---|---|---|---|
| | #Block | #Head | #Block | #Head | #Block | #Head |
| Teacher | 8 | 4 | 8 | 4 | 8 | 8 |
| Student | 1 | 2 | 1 | 1 | 1 | 2 |

Table 7: Overlap rate at the beginning (2% of total training epochs), midpoint (20% of total training epochs), and end (100% of total training epochs) of training. Denoted as OV@2, OV@20, and OV@100 respectively.

| T→S | CiteULike | | | Gowalla | | | Yelp | | |
|---|---|---|---|---|---|---|---|---|---|
| | OR@2 | OR@20 | OR@100 | OR@2 | OR@20 | OR@100 | OR@2 | OR@20 | OR@100 |
| MF→MF | 0.57 | 0.89 | 0.98 | 0.69 | 0.93 | 0.98 | 0.64 | 0.94 | 0.95 |
| LGCN→LGCN | 0.67 | 0.94 | 0.97 | 0.60 | 0.90 | 0.96 | 0.71 | 0.95 | 0.97 |
| HSTU→HSTU | 0.69 | 0.96 | 0.98 | 0.67 | 0.92 | 0.97 | 0.73 | 0.93 | 0.98 |
| LGCN→MF | 0.52 | 0.92 | 0.95 | 0.56 | 0.90 | 0.96 | 0.62 | 0.93 | 0.95 |
| HSTU→MF | 0.54 | 0.88 | 0.97 | 0.61 | 0.92 | 0.95 | 0.67 | 0.95 | 0.96 |

**Teacher/Student.** For each backbone, we create two instances, one large and one small. We use the large instance as the teacher and the small one as the student. For the large instance, we increase the model size until the recommendation performance no longer improves and adopt the model with the best performance. For the small instance, we select the hyperparameters to enlarge the performance gap between the student and the teacher.

Concretely, for MF and LightGCN, we choose different embedding dimensions for the teacher and the student while keeping other hyperparameters the same. The detailed embedding dimensions are provided in Table 5. As for HSTU, we decrease the number of transformer blocks and the number of heads to obtain the student model. The final number of blocks and heads for HSTU is given in Table 6.

In addition to homogeneous settings, we consider two heterogeneous settings where teachers and students have different architectures: 1) LightGCN as the teacher and MF as the student, and 2) HSTU as the teacher and MF as the student.

**Implementation Details.** We implement all the methods with PyTorch and use Adam as the optimizer. Before distillation, we save the teacher's predictions and load them during KD instead of rerunning the teacher. In the case of using HSTU as the student, we fix the batch size to 128. In other cases, we fix it to 2048. For our method, the weight decay is selected from {1e-3, 1e-5, 1e-7}. The search space of the learning rate is {1e-3, 1e-4}. $\beta$ is selected from {0.5,1,3,5,7,9}. $\lambda$ is selected from {0.5,1,5,10,50,100,500,5000,10000}. $K$ and $L$ are both selected from {10,50,100,500,1000}. We conduct early stopping according to the NDCG@20 on the validation set and stop training when the NDCG@20 does not increase for 30 consecutive epochs. All hyperparameters of the compared baselines are tuned to ensure optimal performance.

**Baselines.** We compare our method with the following knowledge distillation methods:
• CD (Lee et al., 2019) samples unobserved items with a ranking-related distribution and uses a point-wise KD loss.
• RRD (Kang et al., 2020) adopts a list-wise loss to maximize the likelihood of the teacher's recommendation list.
• DCD (Lee & Kim, 2021) corrects what the student has failed to predict with a dual correction loss accurately.
• HetComp (Kang et al., 2023) guides the student model by transferring easy-to-hard knwoledge sequences generated from the teacher's trajectories.
• TARec (Zhuang et al., 2025) proposes a teacher-assisted Wasserstein Knowledge Distillation framework that contains two-stage distillation to bridge the gap between the teacher and student.

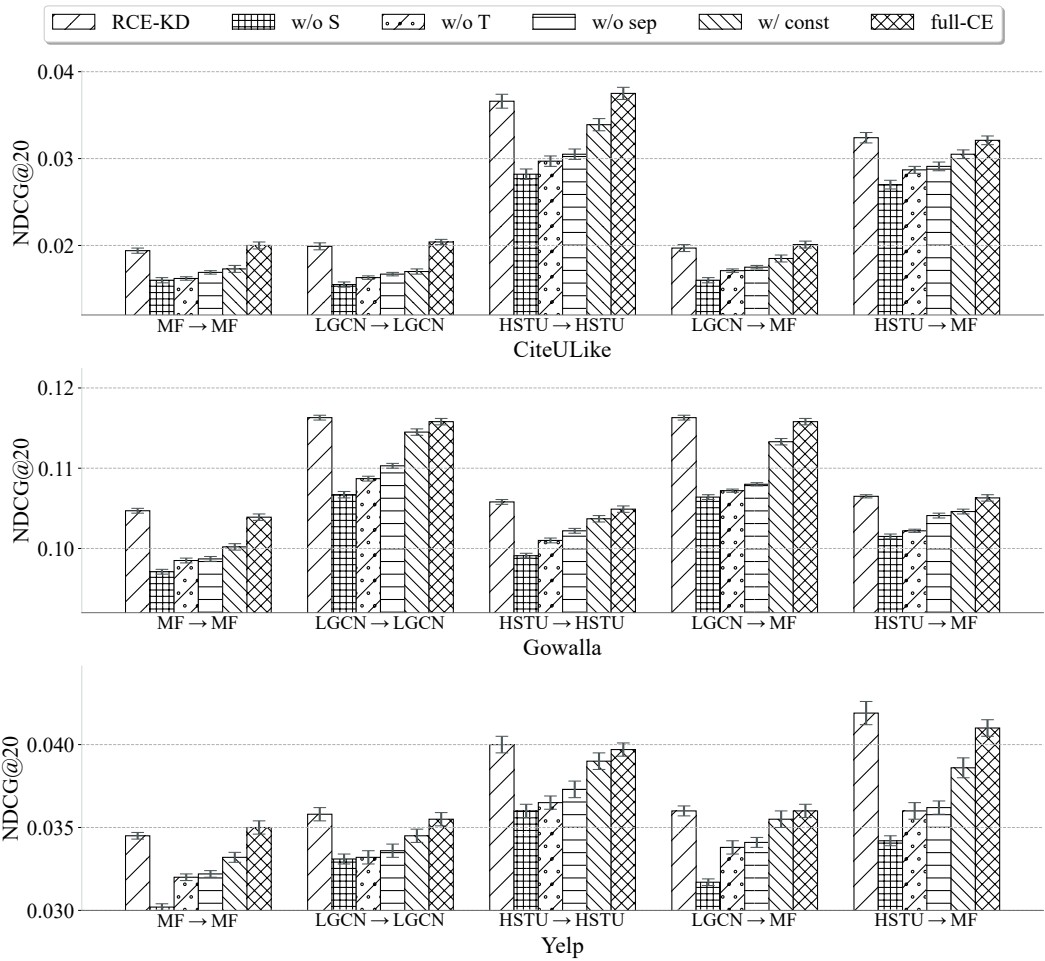

Figure 6: Ablation study on all datasets and all Teacher → Student settings. The average NDCG@20 and standard deviation over 5 independent runs are provided.

## C.2 APPROXIMATE EFFICIENCY OF ASSUMPTION 4.3

In Theorem 4.4, we demonstrate that the relationship between CE loss and NDCG can only be established when Assumption 4.3 holds. To address the practical limitation of precisely satisfying Assumption 4.3 in real-world scenarios, we devise a novel sampling strategy for $(\mathcal{Q}_u^T)_2$ in Section 5.3. This strategy enables the extended set $\mathcal{A}^u$ to closely approximate Assumption 4.3. In this section, we design experiments to validate the efficiency of this approximation. Specifically, we compute the degree of overlap between the set we constructed (i.e., $\mathcal{A}^u$) and the ideal set as training progressed. Formally, we take the top-$|\mathcal{A}^u|$ items given by the student as the ideal set (denoted as $Idea^u$) because it strictly satisfies the closure assumption. Then, in Table 7, we show the overlap rate between $\mathcal{A}^u$ and the ideal set $Idea^u$ at the beginning, midpoint, and end of the training. The overlap rate is computed as $OV = |\mathcal{A}^u \cap Idea^u|/|\mathcal{A}^u \cup Idea^u|$.

From the results in Table 7, we observed that during the early stages of training (approximately 2% of total training epochs), a high overlap rate (exceeding 60%) is typically achieved. As training progresses, the overlap rate increases rapidly, reaching approximately 95% by the mid-training phase (around 20% of total training epochs). By the end of training, the overlap rate reached approximately 98%.

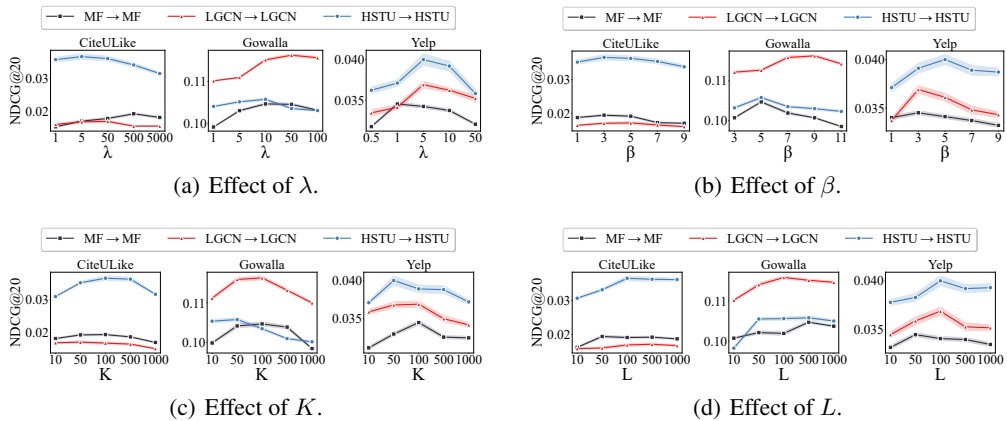

Figure 7: Hyperparameter study on three datasets. We report the results on three homogenous Teacher → Student settings. The average NDCG@20 and standard deviation over 5 independent runs are provided.

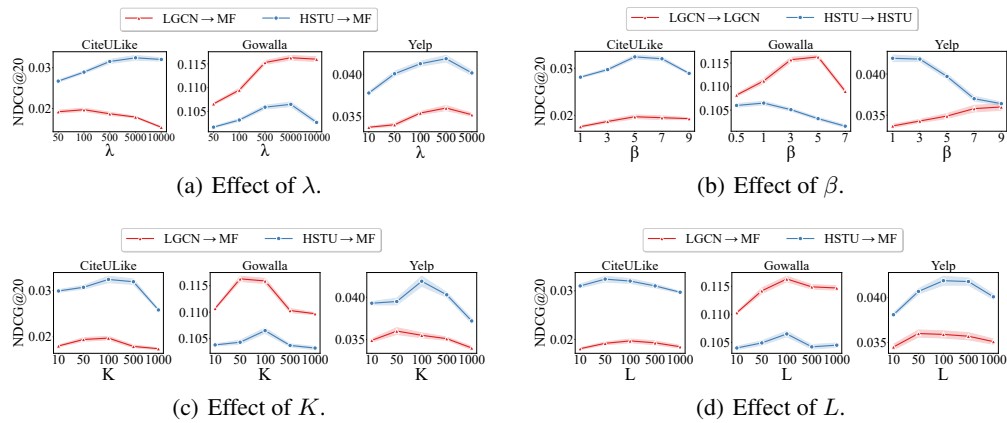

Figure 8: Hyperparameter study on three datasets. We report the results on two heterogeneous Teacher → Student settings. The average NDCG@20 and standard deviation over 5 independent runs are provided.

### C.3 ABLATION STUDY

This section presents additional results of the ablation study. In Figure 6, we give the results on all KD settings and all datasets. The results suggest similar trends to Figure 3. Specifically, we find that all variants are inferior to the original RCE-KD, demonstrating the effectiveness of all key components. Moreover, **RCE-KD w/o S** usually performs worse than **RCE-KD w/o T**. We believe that the reason is that the top items given by the student can exactly satisfy Assumption 4.3, while the top items given by the teacher do not. On the other hand, the superiority of **RCE-KD w/ const** over **RCE-KD w/o T** demonstrates the necessity of involving top items from both the student and the teacher. The superiority of RCE-KD over **RCE-KD w/ const** and **RCE-KD w/o sep** validates the effectiveness of our adaptive weighting scheme and the necessity of splitting out the two subsets and treating them separately. Finally, our method performs even slightly above **full-CE** in most cases, due to a tighter bound on NDCG than full-item CE (Xu et al., 2024a).

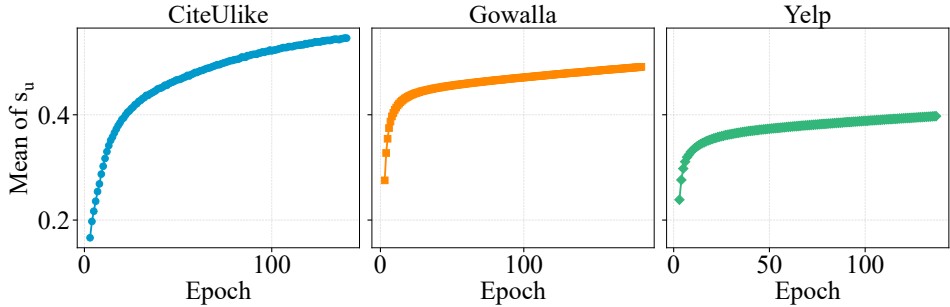

Figure 9: Training curves of the mean of $s_u$ across three datasets (CiteULike, Gowalla, and Yelp).

## C.4 HYPERPARAMETER ANALYSIS

**Effects of $\lambda$.** We use $\lambda$ to balance the impact of our KD loss and the base loss in Eq.(11). In Figure 7(a) and Figure 8(a), we report the effect of $\lambda$. The results suggest that the suitable values of $\lambda$ vary across different datasets. For general, the best choice of $\lambda$ lies in $\{5, 10, 50\}$.

**Effects of $\beta$.** In Eq.(10), we use $\beta$ for computing the adaptive weight. In Figure 7(b) and Figure 8(b), we analyze the effect of $\beta$. The best choice of $\beta$ lies in $\{3, 5, 7\}$. We find that both too large or too small $\beta$ will lead to worse performance because neither of them takes into account both subsets (i.e., $(\mathcal{Q}_u^T)_1$ and $(\mathcal{Q}_u^T)_2$) in the same time.

**Effects of $K$.** We define $\mathcal{Q}_u^T$ and $\mathcal{Q}_u^S$ as the set of items with top-$K$ scores predicted by the teacher and the student, respectively. Here, the hyperparameter $K$ affects the size of these two subsets. In Figure 7(c) and Figure 8(c), we analyze the effect of $K$. We observe that $K$ is optimal at 50 or 100. If $K$ is too small, it will result in key items being ignored; if $K$ is too large, it will introduce too much noise. Thus, choosing a suitable $K$ will benefit the performance.

**Effects of $L$.** When constructing $\mathcal{A}^u$ for the second loss $\mathcal{L}_2$, we sample $L$ items through our proposed sampling strategy. Figure 7(d) and Figure 8(d) analyze the effect of $L$. We find that the optimal value of $L$ is 100. We also find that the performance is less sensitive to the change of $L$ than $K$. However, since a large $L$ inevitably introduces a larger training cost, we suggest choosing a suitable $L$ by considering both the recommendation accuracy and the training inference.

## C.5 JUSTIFICATION OF ADAPTIVELY SCHEDULING $\gamma$

In this section, we justify our $\gamma$-rule through three principals: (1) decreasing, (2) user-specific, and (3) exponential.

### C.5.1 $\mathcal{L}_1$'S WEIGHT MUST INCREASE OVER TRAINING

Our distillation objective is

$$\mathcal{L}_{\text{RCE-KD}} = (1 - \gamma)\mathcal{L}_1 + \gamma \mathcal{L}_2. \tag{34}$$

Let

$$G_1 = \|\nabla \mathcal{L}_1\|, \quad G_2 = \|\nabla \mathcal{L}_2\|. \tag{35}$$

Then the gradient contribution of $\mathcal{L}_1$ is

$$w_1(\gamma) = \frac{(1 - \gamma)G_1}{(1 - \gamma)G_1 + \gamma G_2}, \quad \frac{dw_1}{d\gamma} < 0. \tag{36}$$

Thus decreasing $\gamma$ strictly increases the influence of $\mathcal{L}_1$.

We compare three schedules: 1) fixed $\gamma$, 2) decreasing $\gamma$, and 3) increasing $\gamma$. The results are given in Table 8.

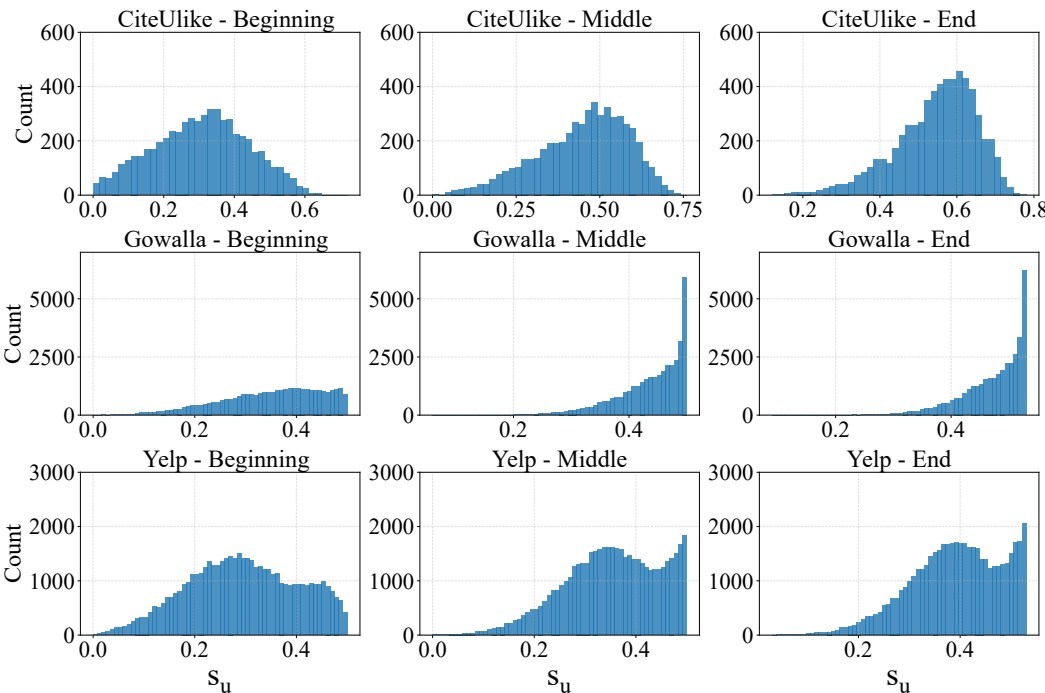

Figure 10: The histogram of $s_u$ at the beginning, middle, and end of training.

Table 8: Recommendation performance of different scheduling methods: 1) fixed $\gamma$, 2) decreasing $\gamma$, and 3) increasing $\gamma$.

| T→S | Schedule | CiteULike | | Gowalla | | Yelp | |
|---|---|---|---|---|---|---|---|
| | | Recall@20 | NDCG@20 | Recall@20 | NDCG@20 | Recall@20 | NDCG@20 |
| MF→MF | fixed $\gamma$ | 0.0393 | 0.0169 | 0.1440 | 0.0996 | 0.0627 | 0.0324 |
| | decreasing $\gamma$ | **0.0410** | **0.0174** | **0.1459** | **0.1003** | **0.0634** | **0.0327** |
| | increasing $\gamma$ | 0.0372 | 0.0164 | 0.1421 | 0.0990 | 0.0624 | 0.0318 |
| LGCN→LGCN | fixed $\gamma$ | 0.0337 | 0.0157 | 0.1630 | 0.1110 | 0.0671 | 0.0347 |
| | decreasing $\gamma$ | **0.0342** | **0.0160** | **0.1639** | **0.1117** | **0.0674** | **0.0350** |
| | increasing $\gamma$ | 0.0335 | 0.0152 | 0.1621 | 0.1106 | 0.0669 | 0.0342 |
| HSTU→HSTU | fixed $\gamma$ | 0.0615 | 0.0330 | 0.1532 | 0.1021 | 0.0724 | 0.0374 |
| | decreasing $\gamma$ | **0.0626** | **0.0341** | **0.1540** | **0.1024** | **0.0726** | **0.0380** |
| | increasing $\gamma$ | 0.0603 | 0.0319 | 0.1527 | 0.1008 | 0.0717 | 0.0366 |

We find that only the schedules that increase $\mathcal{L}_1$-weight with epoch consistently improve NDCG. This matches our principal of amplifying the influence of $\mathcal{L}_1$ and confirms that $\mathcal{L}_1$ should progressively dominate as training proceeds.

### C.5.2 WHY $\gamma$ MUST BE USER-SPECIFIC—AND WHY USING $s_u$ IS JUSTIFIED

In the previous section, $\gamma$ changes with epoch because we analyze global learning dynamics. To enable user-specific adaptivity, however, epoch alone is insufficient: different users progress at different speeds. Thus we define

$$s_u = \frac{|(Q_u^T)_1|}{|Q_u^T|}, \tag{37}$$

which measures how well user $u$ aligns with the teacher.

Figure 9 shows that on average, $s_u$ increases monotonically with training proceeding. This implies that making $\mathcal{L}_1$-weight increase with $s_u$ is consistent with (and indeed equivalent to) the global

Table 9: Performance comparison between global and user-specific scheduling strategies under different teacher–student settings.

| T→S | Schedule | CiteULike | | Gowalla | | Yelp | |
|---|---|---|---|---|---|---|---|
| | | Recall@20 | NDCG@20 | Recall@20 | NDCG@20 | Recall@20 | NDCG@20 |
| MF→MF | Global | 0.0410 | 0.0174 | 0.1459 | 0.1003 | 0.0634 | 0.0327 |
| | User-specific | **0.0419** | **0.0181** | **0.1477** | **0.1012** | **0.0648** | **0.0333** |
| LGCN→LGCN | Global | 0.0342 | 0.0160 | 0.1639 | 0.1117 | 0.0674 | 0.0350 |
| | User-specific | **0.0363** | **0.0164** | **0.1664** | **0.1133** | **0.0699** | **0.0357** |
| HSTU→HSTU | Global | 0.0626 | 0.0341 | 0.1540 | 0.1024 | 0.0726 | 0.0380 |
| | User-specific | **0.0643** | **0.0350** | **0.1567** | **0.1032** | **0.0734** | **0.0388** |

Table 10: Performance comparison between linear and exponential scheduling strategies under different teacher–student settings.

| T→S | Schedule | CiteULike | | Gowalla | | Yelp | |
|---|---|---|---|---|---|---|---|
| | | Recall@20 | NDCG@20 | Recall@20 | NDCG@20 | Recall@20 | NDCG@20 |
| MF→MF | Linear | 0.0419 | 0.0181 | 0.1477 | 0.1012 | 0.0648 | 0.0333 |
| | Exponential | **0.0431** | **0.0194** | **0.1525** | **0.1047** | **0.0667** | **0.0345** |
| LGCN→LGCN | Linear | 0.0363 | 0.0164 | 0.1664 | 0.1133 | 0.0699 | 0.0357 |
| | Exponential | **0.0377** | **0.0171** | **0.1681** | **0.1163** | **0.0716** | **0.0369** |
| HSTU→HSTU | Linear | 0.0643 | 0.0350 | 0.1567 | 0.1032 | 0.0734 | 0.0388 |
| | Exponential | **0.0670** | **0.0366** | **0.1594** | **0.1058** | **0.0754** | **0.0400** |

decision that $\mathcal{L}_1$-weight should increase with training proceeding. The crucial difference is that $s_u$ captures heterogeneous per-user progress, whereas epoch does not.

Because the distribution of $\{s_u\}$ remains highly dispersed during training (as shown in Figure 10), hard users retain low $s_u$ while easy users obtain high $s_u$. A global $\gamma$ forces both types of users to adopt the same $\mathcal{L}_1/\mathcal{L}_2$ weight ratio, which is suboptimal. Thus we adopt the user-specific schedule

$$\gamma_u = as_u + b, \quad \text{L1-weight} = 1 - \gamma_u. \tag{38}$$

Table 9 shows that this user-specific rule significantly outperforms global $\gamma$.

### C.5.3 WHY THE INCREASE SHOULD BE EXPONENTIAL

Having established a monotonic increase in $\mathcal{L}_1$-weight, we compare two increasing schedules:

- **Linear**: $1 - \gamma_{\text{lin}}(s_u) = as_u + b$
- **Exponential**: $1 - \gamma_{\text{exp}}(s_u) = 1 - e^{-\beta s_u}$

The exponential form provides a phase transition, yielding low weight of $\mathcal{L}_1$ initially (allowing $\mathcal{L}_2$ to explore) and rapid change later (letting $\mathcal{L}_1$ dominate).

Table 10 shows that the exponential rule consistently outperforms the linear rule, validating its principled nature.

### C.5.4 FINAL JUSTIFICATION

From the analyses above, a proper schedule must:

1. Increase $\mathcal{L}_1$-weight over the course of training.
2. Adapt per user, which requires using $s_u$.
3. Follow an exponential pattern.

Our final rule is the minimal functional form that satisfies all three principles. Alternative schedules violate at least one of these principles and perform worse.

Table 11: The Number of transformer blocks (#Block) and number of heads (#Head) for SASRec.

| Model | CiteULike | | Gowalla | | Yelp | |
|---|---|---|---|---|---|---|
| | #Block | #Head | #Block | #Head | #Block | #Head |
| Teacher | 4 | 2 | 2 | 4 | 8 | 4 |
| Student | 1 | 1 | 1 | 1 | 1 | 2 |

## D  APPLICATION ON MORE RECOMMENDATION TASKS

### D.1  APPLICABILITY IN SEQUENTIAL RECOMMENDATION

Our method can be easily applied to other recommendation scenarios, such as sequential recommendation. To verify this, we construct sequential recommendation datasets using the datasets in our paper. Specifically, we take each user's last interaction as the test item, the second-to-last interaction as the validation item, and the previous interactions as training items. To further enrich our evaluation, we introduce the classical sequential recommendation model SASRec (Kang & McAuley, 2018) as an additional backbone. We include a new distillation setting in which both the teacher and the student are based on SASRec. The detailed configurations for this setup are provided in Table 11.

In Table 12, we report the performance of all methods under five knowledge distillation settings. The results demonstrate that our approach still significantly outperforms all baseline methods on sequence recommendation tasks. Compared to the best baseline method, our method achieves improvements ranging from a minimum of 2.74% to a maximum of 10%. This performance is comparable to our results on top-N recommendation tasks presented in the main text, indicating the strong generalization capability of our method across recommendation tasks.

### D.2  APPLICABILITY IN MULTI-MODAL RECOMMENDATION

To further examine whether our approach generalizes to multi-modal recommendation scenarios, we conduct experiments on three widely used multi-modal recommendation datasets:

- **Netflix**: The Netflix dataset (Wei et al., 2024) provides user–item interaction logs collected from the streaming platform. Each movie is accompanied by multimodal metadata, most notably poster images and titles. Visual representations are derived using the CLIP-ViT encoder (Radford et al., 2021), while textual information is transformed into embeddings via a pre-trained BERT model (Devlin et al., 2019).

- **Tiktok**: The Tiktok micro-video dataset (Wei et al., 2023) offers interaction histories together with three complementary types of content: visual, acoustic, and textual. Visual and audio signals from short videos are processed into 128-dimensional feature vectors, and the accompanying captions are embedded using Sentence-BERT (Reimers & Gurevych, 2019).

- **Electronics**: The Amazon Electronics dataset (He & McAuley, 2016; McAuley et al., 2015) consists of user reviews and item metadata from the consumer electronics category. Images are represented using 4,096-dimensional features extracted by pre-trained CNN-based models (He & McAuley, 2016). Textual information—formed by integrating titles, descriptions, category labels, and brand attributes—is encoded as 384-dimensional embeddings using Sentence-BERT (Reimers & Gurevych, 2019).

In this task of knowledge distillation for multi-modal recommender systems, we compare our method with three baselines. Specifically, we select two state-of-the-art knowledge distillation methods for multi-modal recommender systems: **PromptMM** (Wei et al., 2024) and **TARec** (Zhuang et al., 2025). PromptMM introduces soft prompt-tuning, together with a disentangled multi-modal list-wise distillation and modality-aware re-weighting mechanism. TARec proposes a teacher-assisted Wasserstein Knowledge Distillation framework that contains two-stage distillation to bridge the gap between the teacher and student. We also utilize **HetComp** (Kang et al., 2023), the state-of-the-art knowledge distillation method for general recommender systems as a baseline. Note that our method and HetComp rely solely on the teacher's logits for distillation, whereas PromptMM and TARec use both logits and embeddings. To ensure a fair comparison, we disable their embedding-level distillation components and retain all other parts of their methods.

Table 12: Recommendation performance on sequential recommendation task. The best results are in boldface, and the best baselines are underlined. *Improv.b* denotes the relative improvement of RCE-KD over the best baseline. LGCN stands for LightGCN. A paired t-test is performed over 5 independent runs for evaluating $p$-value ($\leq 0.05$ indicates statistical significance).

| T→S | Method | CiteULike | | Gowalla | | Yelp | |
|---|---|---|---|---|---|---|---|
| | | Recall@10 | NDCG@10 | Recall@10 | NDCG@10 | Recall@10 | NDCG@10 |
| MF→MF | Teacher | 0.0077 | 0.0063 | 0.0310 | 0.0208 | 0.0153 | 0.0080 |
| | Student | 0.0046 | 0.0032 | 0.0161 | 0.0081 | 0.0097 | 0.0058 |
| | CD | 0.0067 | 0.0054 | 0.0274 | 0.0175 | 0.0140 | 0.0071 |
| | RRD | 0.0069 | 0.0057 | 0.0281 | 0.0182 | 0.0139 | 0.0069 |
| | DCD | 0.0073 | 0.0060 | 0.0292 | 0.0189 | 0.0142 | 0.0071 |
| | HetComp | 0.0072 | 0.0059 | 0.0289 | 0.0187 | 0.0146 | 0.0074 |
| | TARec | 0.0070 | 0.0056 | 0.0285 | 0.0184 | 0.0140 | 0.0071 |
| | RCE-KD | **0.0078** | **0.0063** | **0.0305** | **0.0201** | **0.0150** | **0.0077** |
| | *Improv.b* | 6.85% | 5.00% | 4.45% | 6.35% | 2.74% | 4.05% |
| | *p-value* | 3.9e-4 | 8.6e-4 | 5.5e-3 | 3.7e-4 | 6.7e-3 | 7.7e-4 |
| LGCN→LGCN | Teacher | 0.0083 | 0.0066 | 0.0401 | 0.0279 | 0.0167 | 0.0089 |
| | Student | 0.0051 | 0.0040 | 0.0217 | 0.0154 | 0.0103 | 0.0064 |
| | CD | 0.0066 | 0.0050 | 0.0349 | 0.0250 | 0.0144 | 0.0073 |
| | RRD | 0.0068 | 0.0051 | 0.0354 | 0.0251 | 0.0146 | 0.0077 |
| | DCD | 0.0070 | 0.0055 | 0.0368 | 0.0261 | 0.0154 | 0.0082 |
| | HetComp | 0.0071 | 0.0055 | 0.0363 | 0.0259 | 0.0148 | 0.0079 |
| | TARec | 0.0068 | 0.0052 | 0.0366 | 0.0261 | 0.0157 | 0.0083 |
| | RCE-KD | **0.0075** | **0.0060** | **0.0383** | **0.0269** | **0.0165** | **0.0087** |
| | *Improv.b* | 5.63% | 9.09% | 4.08% | 3.07% | 5.10% | 4.82% |
| | *p-value* | 9.6e-5 | 9.0e-4 | 3.1e-4 | 8.7e-4 | 7.2e-4 | 3.8e-4 |
| SASRec→SASRec | Teacher | 0.0091 | 0.0067 | 0.0325 | 0.0214 | 0.0159 | 0.0087 |
| | Student | 0.0054 | 0.0042 | 0.0160 | 0.0086 | 0.0100 | 0.0067 |
| | CD | 0.0073 | 0.0055 | 0.0250 | 0.0169 | 0.0133 | 0.0070 |
| | RRD | 0.0079 | 0.0057 | 0.0241 | 0.0165 | 0.0129 | 0.0069 |
| | DCD | 0.0083 | 0.0064 | 0.0269 | 0.0170 | 0.0143 | 0.0074 |
| | HetComp | 0.0086 | 0.0066 | 0.0281 | 0.0174 | 0.0136 | 0.0070 |
| | TARec | 0.0080 | 0.0059 | 0.0271 | 0.0171 | 0.0130 | 0.0067 |
| | RCE-KD | **0.0092** | **0.0070** | **0.0303** | **0.0181** | **0.0154** | **0.0078** |
| | *Improv.b* | 6.98% | 6.06% | 7.83% | 4.02% | 7.69% | 5/41% |
| | *p-value* | 1.3e-3 | 3.7e-4 | 6.7e-4 | 5.5e-5 | 3.9e-3 | 9.1e-4 |
| HSTU→HSTU | Teacher | 0.0102 | 0.0072 | 0.0331 | 0.0217 | 0.0172 | 0.0098 |
| | Student | 0.0063 | 0.0049 | 0.0164 | 0.0087 | 0.0110 | 0.0071 |
| | CD | 0.0102 | 0.0067 | 0.0285 | 0.0151 | 0.0129 | 0.0075 |
| | RRD | 0.0092 | 0.0065 | 0.0270 | 0.0157 | 0.0134 | 0.0078 |
| | DCD | 0.0095 | 0.0069 | 0.0277 | 0.0162 | 0.0141 | 0.0082 |
| | HetComp | 0.0099 | 0.0073 | 0.0282 | 0.0170 | 0.0149 | 0.0084 |
| | TARec | 0.0093 | 0.0067 | 0.0271 | 0.0157 | 0.0140 | 0.0081 |
| | RCE-KD | **0.0111** | **0.0078** | **0.0309** | **0.0184** | **0.0158** | **0.0090** |
| | *Improv.b* | 8.82% | 6.85% | 8.42% | 8.24% | 6.04% | 7.14% |
| | *p-value* | 5.7e-4 | 8.9e-4 | 9.0e-5 | 2.0e-3 | 9.7e-4 | 4.1e-3 |
| HSTU→MF | Teacher | 0.0102 | 0.0072 | 0.0331 | 0.0217 | 0.0172 | 0.0098 |
| | Student | 0.0046 | 0.0032 | 0.0161 | 0.0081 | 0.0097 | 0.0058 |
| | CD | 0.0070 | 0.0053 | 0.0287 | 0.0188 | 0.0153 | 0.0079 |
| | RRD | 0.0077 | 0.0059 | 0.0301 | 0.0200 | 0.0160 | 0.0091 |
| | DCD | 0.0082 | 0.0061 | 0.0303 | 0.0200 | 0.0164 | 0.0091 |
| | HetComp | 0.0089 | 0.0065 | 0.0310 | 0.0207 | 0.0163 | 0.0089 |
| | TARec | 0.0083 | 0.0062 | 0.0298 | 0.0197 | 0.0164 | 0.0093 |
| | RCE-KD | **0.0097** | **0.0068** | **0.0329** | **0.0214** | **0.0173** | **0.0098** |
| | *Improv.b* | 8.99% | 4.62% | 6.13% | 3.38% | 5.49% | 5.38% |
| | *p-value* | 4.2e-4 | 9.3e-3 | 5.3e-4 | 9.2e-5 | 6.2e-3 | 7.4e-4 |

We treat the multi-modal teacher model BM3 (Zhou et al., 2023) as teacher backbones since it performs well in recent work (Zhuang et al., 2025). Per the custom of recent work (Wei et al., 2024; Zhuang et al., 2025), we use the ID-based models MF and LightGCN as students. This yields our teacher→student configurations: **BM3→MF, BM3→LGCN**.

Table 15 summarizes the experimental results. In all settings, RCE-KD significantly improves over the students and consistently outperforms all baseline methods. In most cases, our method approaches the teacher's performance, remaining just slightly lower, or even slightly higher, while other methods are farther behind. These results confirm that our distillation approach remains effective when

Table 13: Statistics of datasets with multi-modal item Visual (V), Acoustic (A), Textual (T) contents.

| Dataset | Netflix | | Tiktok | | | Electronics | |
|---|---|---|---|---|---|---|---|
| Modality | V | T | V | A | T | V | T |
| Feat. Dim. | 512 | 768 | 128 | 128 | 768 | 4096 | 384 |
| User | 43,739 | | 14,343 | | | 41,691 | |
| Item | 17,239 | | 8,690 | | | 21,479 | |
| Interaction | 609,341 | | 276,637 | | | 359,165 | |
| Sparsity | 99.919% | | 99.778% | | | 99.960% | |

Table 14: Dimensions of models on multi-modal recommendation task.

| Dataset | Netflix | | | Tiktok | | | Electronics | | |
|---|---|---|---|---|---|---|---|---|---|
| Model | BM3 | MF | LightGCN | BM3 | MF | LightGCN | BM3 | MF | LightGCN |
| Dimension | 400 | 20 | 20 | 200 | 20 | 20 | 200 | 20 | 20 |

Table 15: Recommendation performance on multi-modal recommendation task. The best results are in boldface, and the best baselines are underlined. *Improv.b* denotes the relative improvement of RCE-KD over the best baseline. LGCN stands for LightGCN. A paired t-test is performed over 5 independent runs for evaluating $p$-value ($\leq 0.05$ indicates statistical significance).

| T→S | Method | Netflix Recall@20 | Netflix NDCG@20 | Tiktok Recall@20 | Tiktok NDCG@20 | Electronics Recall@20 | Electronics NDCG@20 |
|---|---|---|---|---|---|---|---|
| | Teacher | 0.1913 | 0.0809 | 0.0707 | 0.0254 | 0.0361 | 0.0152 |
| | Student | 0.1420 | 0.0498 | 0.0417 | 0.0153 | 0.0182 | 0.0073 |
| BM3→MF | HetComp | 0.1783 | 0.0755 | 0.0659 | 0.0238 | 0.0320 | 0.0133 |
| | PromptMM | 0.1732 | 0.0728 | 0.0637 | 0.0221 | 0.0297 | 0.0120 |
| | TARec | 0.1792 | 0.0760 | 0.0656 | 0.0230 | 0.0333 | 0.0139 |
| | RCE-KD | **0.1884** | **0.0797** | **0.0702** | **0.0253** | **0.0347** | **0.0146** |
| | *Improv.b* | 5.13% | 4.87% | 6.53% | 6.30% | 4.20% | 5.04% |
| | *p-value* | 2.0e-3 | 5.9e-4 | 1.7e-3 | 9.7e-5 | 6.6e-3 | 2.8e-4 |
| | Teacher | 0.1913 | 0.0809 | 0.0707 | 0.0254 | 0.0361 | 0.0152 |
| | Student | 0.1507 | 0.0537 | 0.0562 | 0.0189 | 0.0211 | 0.083 |
| BM3→LGCN | HetComp | 0.1811 | 0.0747 | 0.0629 | 0.0220 | 0.0317 | 0.0129 |
| | PromptMM | 0.1779 | 0.0728 | 0.0603 | 0.0215 | 0.0323 | 0.0137 |
| | TARec | 0.1834 | 0.0753 | 0.0651 | 0.0235 | 0.0341 | 0.0144 |
| | RCE-KD | **0.1907** | **0.0804** | **0.0705** | **0.0255** | **0.0359** | **0.0153** |
| | *Improv.b* | 3.98% | 6.77% | 8.29% | 8.51% | 5.28% | 6.25% |
| | *p-value* | 7.9e-4 | 4.0e-3 | 9.4e-4 | 5.9e-3 | 7.3e-4 | 8.2e-5 |

transferring knowledge from multi-modal teachers to lighter ID-based students. Our experimental results demonstrate that the proposed method is not limited to distillation within ID-based models.

