# OpenReview forum: "Rejuvenating Cross-Entropy Loss in Knowledge Distillation for Recommender Systems"
_ICLR.cc/2026/Conference — ICLR 2026 Poster_

### Official Review · Reviewer_irWM · 2025-10-25

**Soundness:** 3
**Presentation:** 3
**Contribution:** 3
**Rating:** 6
**Confidence:** 3

**Summary:**

This paper proposed an improved version of cross entropy loss for knowledge distillation

**Strengths:**

1. The theoretical claims seem interesting.
2. The presentation of the paper is clear.
3. Experimental results demonstrate that the proposed method can show performance improvement.

**Weaknesses:**

1. It will be better if more baselines specialized for knowledge distillation in recommender systems can be compared such as [1][2]. However, I think the paper be of interest even without these comparisons, and the comparison will strengthen the claims.
2. The design of gamma update rule in Eq 10 seems somewhat ad-hoc. Maybe more principled method can be developed to balance this weight parameter.
3. The theory part is interesting but might not be entirely novel. The fact that optimizing recommendation loss function is connected to the ranking metric NCDG [3] is relatively known in the community, and in my opinion the majority of knowledge distillation loss derivation in this paper is centered around this.

References

[1] PromptMM: Multi-Modal Knowledge Distillation for Recommendation with Prompt-Tuning

[2] Unbiased knowledge distillation for recommendation

[3] On Optimizing Top-𝐾 Metrics for Neural Ranking Models

**Questions:**

Can the proposed knowledge distillation method be applied to multi-modal recommender systems or just recommender system without modality features?

---

> ### Author Response · Authors · 2025-11-21
> **Response to Weakness 1 & 2**
>
> Thank you very much for your recognition of our work!
>
> > **Weakness 1:** It will be better if more baselines specialized for knowledge distillation in recommender systems can be compared such as [1,2].
>
> Thank you very much for your suggestion. Since PromptMM requires item-level multimodal information, we are unable to include it in the main experimental results (Table 1). However, **we have incorporated PromptMM as one of the baselines in our newly added multimodal experiments (Appendix D.2)**. **We also include a recently proposed distillation method TARec as a new baseline in all of our experiments.** Compared with PromptMM and TARec, UnKD is relatively outdated, so we did not include it in our experiments. Although these new baselines are quite strong, they still fall short of the performance achieved by our method.
>
> The comparison results with TARec in top-N recommendation in our main text is given below:
>
> | T→S           | Method | CiteULike  |            | Gowalla    |            | Yelp       |            |
> | ------------- | ------ | ---------- | ---------- | ---------- | ---------- | ---------- | ---------- |
> |               |        | R@20       | N@20       | R@20       | N@20       | R@20       | N@20       |
> | **MF→MF**     | TARec  | 0.0380     | 0.0171     | 0.1416     | 0.0993     | 0.0635     | 0.0327     |
> |               | RCE-KD | **0.0431** | **0.0194** | **0.1525** | **0.1047** | **0.0667** | **0.0345** |
> | **LGCN→LGCN** | TARec  | 0.0350     | 0.0157     | 0.1630     | 0.1112     | 0.0694     | 0.0352     |
> |               | RCE-KD | **0.0377** | **0.0171** | **0.1681** | **0.1163** | **0.0716** | **0.0369** |
> | **HSTU→HSTU** | TARec  | 0.0571     | 0.0299     | 0.1537     | 0.1005     | 0.0713     | 0.0369     |
> |               | RCE-KD | **0.0670** | **0.0366** | **0.1594** | **0.1058** | **0.0754** | **0.0400** |
> | **HSTU→MF**   | TARec  | 0.0370     | 0.0173     | 0.1603     | 0.1100     | 0.0663     | 0.0350     |
> |               | RCE-KD | **0.0437** | **0.0197** | **0.1677** | **0.1163** | **0.0692** | **0.0360** |
> | **LGCN→MF**   | TARec  | 0.0538     | 0.0300     | 0.1527     | 0.1023     | 0.0762     | 0.0378     |
> |               | RCE-KD | **0.0595** | **0.0324** | **0.1624** | **0.1065** | **0.0805** | **0.0419** |
>
> For complete results, please refer to **Table 1** in our updated submission. For results in sequential recommendation, please refer to **Table 9**. For results in multi-modal recommendation and comparison with PrompMM, please refer to **Table 12**.
>
> > **Weakness 2:** The design of gamma update rule in Eq 10 seems somewhat ad-hoc.
>
> The update rule for $\\gamma$ is not ad-hoc; rather, **it is deliberately designed to reflect the distinct roles of $\mathcal{L}\_1$ and $\mathcal{L}\_2$ in our framework**. The loss $\mathcal{L}\_1$ strictly satisfies Assumption 4.3, and optimizing it provides the most direct theoretical guarantee for improving NDCG. In contrast, $\mathcal{L}\_2$ serves a supporting purpose: it enlarges the first item set $(\mathcal Q\_u^T)\_1$, ensuring that more items can fall under the influence of $\mathcal{L}\_1$, and it also supplements when contributions from $\mathcal{L}\_1$ are insufficient.
>
> To achieve this balance, we adopt an **exponential schedule** for updating $\\gamma$. This design produces **a controlled “jump” behavior**: once the size of $(\mathcal Q\_u^T)\_1$ reaches a desired threshold, the weight of $\mathcal{L}\_1$ increases rapidly. This ensures that the strictly theoretically grounded component $\mathcal{L}\_1$ quickly becomes dominant at the appropriate stage of training, while still allowing $\mathcal{L}\_2$ to perform its supporting role since its weight is not zero.
>
> In this sense, the update rule is principled and purpose-driven—shaped by the theoretical properties of our loss components.
>
> Thank you for bringing this to our attention. **We have incorporated the aforementioned discussion into the blue-highlighted section of Section 5.4**.

---

> > ### Comment · Reviewer_irWM · 2025-11-24
> >
> > For weakness 2 it seems that no blue-highlighted text is updated in the main paper. Also, to demonstrate your update method is really "principled", it will be better if you may conduct some gradient analysis or other analysis to show that the proposed loss function can adaptively balance between L1 and L2, and other scheduling methods may not achieve this desired effect. The theory part points how the necessity of L1 and L2, but it does not state how L1 and L2 should be combined. And in the current version, there seem to be few evidence demonstrating that this combination method (the update rule in Eq 10) is justified.

---

> > > ### Author Response · Authors · 2025-11-25
> > > **Response to scheduling $\gamma$ (1)**
> > >
> > > We sincerely apologize for forgetting to highlight the text in blue. The following content has been incorporated into the revised manuscript and included as **Appendix C.5**.
> > >
> > > We clarify that our $\gamma$-update rule is **not** designed to “balance’’ $\mathcal{L}_1$ and $\mathcal{L}_2$. The two losses are asymmetric in nature, as demonstrated in our recent rebuttal: $\mathcal{L}_1$ strictly satisfies Assumption 4.3, and optimizing it provides the most direct theoretical guarantee for improving NDCG. In contrast, $\mathcal{L}_2$ serves a supporting purpose: it enlarges the first item set $(\mathcal Q_u^T)_1$, ensuring that more items can fall under the influence of $\mathcal{L}_1$, and it also supplements when contributions from $\mathcal{L}_1$ are insufficient.
> > >
> > > Therefore, our goal is **not** to keep them in equilibrium but to **progressively amplify the influence of $\mathcal{L}_1$**. Below **we justify our $\gamma$-rule through three principals: (1) decreasing, (2) user-specific, and (3) exponential**.
> > >
> > > ## (1) $\mathcal L_1$’s weight must increase over training
> > >
> > > Our distillation objective is
> > > $$
> > > \mathcal{L}_{\text{RCE-KD}}
> > > = (1-\gamma)\mathcal{L}_1 + \gamma\mathcal{L}_2.
> > > $$
> > > Let
> > > $$
> > > G_1=\|\nabla\mathcal{L}_1\|,\quad G_2=\|\nabla\mathcal{L}_2\|.
> > > $$
> > > Then the gradient contribution of $\mathcal{L}_1$ is
> > > $$
> > > w_1(\gamma)=\frac{(1-\gamma)G_1}{(1-\gamma)G_1+\gamma G_2},
> > > \quad
> > > \frac{d w_1}{d\gamma}<0.
> > > $$
> > > Thus **decreasing $\gamma$ strictly increases the influence of $\mathcal{L}_1$**.
> > >
> > > We compare three schedules—fixed $\gamma$, decreasing $\gamma$, and increasing $\gamma$. The results are given below (also in Table 8 in our revised manuscript):
> > >
> > > | T→S           | Schedule            | CiteULike  |            | Gowalla    |            | Yelp       |            |
> > > | ------------- | ------------------- | ---------- | ---------- | ---------- | ---------- | ---------- | ---------- |
> > > |               |                     | R@20       | N@20       | R@20       | N@20       | R@20       | N@20       |
> > > | **MF→MF**     | fixed $\gamma$      | 0.0393     | 0.0169     | 0.1440     | 0.0996     | 0.0627     | 0.0324     |
> > > |               | decreasing $\gamma$ | **0.0410** | **0.0174** | **0.1459** | **0.1003** | **0.0634** | **0.0327** |
> > > |               | increasing $\gamma$ | 0.0372     | 0.0164     | 0.1421     | 0.0990     | 0.0624     | 0.0318     |
> > > | **LGCN→LGCN** | fixed $\gamma$      | 0.0337     | 0.0157     | 0.1630     | 0.1110     | 0.0671     | 0.0347     |
> > > |               | decreasing $\gamma$ | **0.0342** | **0.0160** | **0.1639** | **0.1117** | **0.0674** | **0.0350** |
> > > |               | increasing $\gamma$ | 0.0335     | 0.0152     | 0.1621     | 0.1106     | 0.0669     | 0.0342     |
> > > | **HSTU→HSTU** | fixed $\gamma$      | 0.0615     | 0.0330     | 0.1532     | 0.1021     | 0.0724     | 0.0374     |
> > > |               | decreasing $\gamma$ | **0.0626** | **0.0341** | **0.1540** | **0.1024** | **0.0726** | **0.0380** |
> > > |               | increasing $\gamma$ | 0.0603     | 0.0319     | 0.1527     | 0.1008     | 0.0717     | 0.0366     |
> > >
> > > We find that only the schedules that **increase $\mathcal L_1$-weight with epoch** consistently improve NDCG. This matches our principal of amplifying the influence of $\mathcal{L}_1$ and confirms that $\mathcal{L}_1$ should progressively dominate as training proceeds.

---

> > > ### Author Response · Authors · 2025-11-25
> > > **Response to scheduling $\gamma$ (2)**
> > >
> > > ## (2) Why $\gamma$ must be user-specific—and why using $s_u$ is justified
> > >
> > > In (1), $\gamma$ changes **with epoch** because we analyze global learning dynamics.
> > >  To enable *user-specific adaptivity*, however, epoch alone is insufficient: different users progress at different speeds.
> > >  Thus we define
> > > $$
> > > s_u = \frac{|(Q_u^T)_1|}{|Q_u^T|},
> > > $$
> > > which measures how well user $u$ aligns with the teacher.
> > >
> > > **Figure 9 in our revised manuscript shows that on average, $s_u$ increases monotonically with training proceeding.**
> > >
> > > This implies that **making $\mathcal L_1$-weight increase with $s_u$** is consistent with (and indeed equivalent to) the global decision that $\mathcal L_1$-weight should increase with training proceeding.
> > >
> > > The crucial difference is that **$s_u$ captures heterogeneous per-user progress**, whereas epoch does not.
> > >
> > > Because the distribution of $\{s_u\}$ remains highly dispersed during training (as shown in Figure 10 in our revised manuscript), hard users retain low $s_u$ while easy users obtain high $s_u$. A global $\gamma$ forces both types of users to adopt the same $\mathcal L_1$/$\mathcal L_2$ weight ratio, which is suboptimal.
> > >
> > > Thus we adopt the user-specific schedule
> > > $$
> > > \gamma_u = a s_u + b,\text{ where }a<0,
> > > \quad \quad
> > > \text{L1-weight} = 1-\gamma_u.
> > > $$
> > > This design gives easy users (large $s_u$) earlier access to the NDCG-consistent term $\mathcal{L}_1$, and vice versa.
> > >
> > > The table below (also Table 9 in our revised manuscript) shows that this user-specific rule significantly outperforms global $\gamma$.
> > >
> > > | T→S           | Schedule      | CiteULike  |            | Gowalla    |            | Yelp       |            |
> > > | ------------- | ------------- | ---------- | ---------- | ---------- | ---------- | ---------- | ---------- |
> > > |               |               | R@20       | N@20       | R@20       | N@20       | R@20       | N@20       |
> > > | **MF→MF**     | Global        | 0.0410     | 0.0174     | 0.1459     | 0.1003     | 0.0634     | 0.0327     |
> > > |               | User-specific | **0.0419** | **0.0181** | **0.1477** | **0.1012** | **0.0648** | **0.0333** |
> > > | **LGCN→LGCN** | Global        | 0.0342     | 0.0160     | 0.1639     | 0.1117     | 0.0674     | 0.0350     |
> > > |               | User-specific | **0.0363** | **0.0164** | **0.1664** | **0.1133** | **0.0699** | **0.0357** |
> > > | **HSTU→HSTU** | Global        | 0.0626     | 0.0341     | 0.1540     | 0.1024     | 0.0726     | 0.0380     |
> > > |               | User-specific | **0.0643** | **0.0350** | **0.1567** | **0.1032** | **0.0734** | **0.0388** |
> > >
> > > ## (3) Why the increase should be exponential
> > >
> > > Having established a monotonic increase in $\mathcal L_1$-weight, we compare two increasing schedules:
> > >
> > > - **Linear:** $1-\gamma_{\text{lin}}(s_u)=a s_u + b,\text{ where }a<0,$
> > > - **Exponential:** $1-\gamma_{\exp}(s_u)=1-e^{-\beta s_u}$
> > >
> > > The exponential form provides a **phase transition**, yielding low weight of $\mathcal L_1$ initially  (allowing $\mathcal{L}_2$ to explore) and rapid change later (letting $\mathcal{L}_1$ dominate).
> > >
> > > The table below (also Table 10 in our revised manuscript) shows the exponential rule consistently outperforms the linear rule, validating its principled nature.
> > >
> > > | T→S           | Schedule    | CiteULike  |            | Gowalla    |            | Yelp       |            |
> > > | ------------- | ----------- | ---------- | ---------- | ---------- | ---------- | ---------- | ---------- |
> > > |               |             | R@20       | N@20       | R@20       | N@20       | R@20       | N@20       |
> > > | **MF→MF**     | Linear      | 0.0419     | 0.0181     | 0.1477     | 0.1012     | 0.0648     | 0.0333     |
> > > |               | Exponential | **0.0431** | **0.0194** | **0.1525** | **0.1047** | **0.0667** | **0.0345** |
> > > | **LGCN→LGCN** | Linear      | 0.0363     | 0.0164     | 0.1664     | 0.1133     | 0.0699     | 0.0357     |
> > > |               | Exponential | **0.0377** | **0.0171** | **0.1681** | **0.1163** | **0.0716** | **0.0369** |
> > > | **HSTU→HSTU** | Linear      | 0.0643     | 0.0350     | 0.1567     | 0.1032     | 0.0734     | 0.0388     |
> > > |               | Exponential | **0.0670** | **0.0366** | **0.1594** | **0.1058** | **0.0754** | **0.0400** |
> > >
> > > ## Final justification
> > >
> > > From the analyses above, a principled schedule must:
> > >
> > > 1. **Increase $\mathcal L_1$-weight over the course of training**.
> > >
> > > 2. **Adapt per user**, which requires using $s_u$ and is justified because
> > >    $$
> > >    s_u \uparrow \text{ with epoch } \quad \text{and}\quad  \text{captures user-level difficulty}.
> > >    $$
> > >
> > > 3. **Follow an exponential pattern**.
> > >
> > > Our final rule is the **minimal functional form** that satisfies all three principles. Alternative schedules violate at least one of these principles and perform worse, demonstrating that Eq.(10) is principled rather than ad-hoc.

---

> > > > ### Comment · Reviewer_irWM · 2025-11-25
> > > >
> > > > Thanks for your response, which addresses my concern. I will increase my score.

---

> > > > > ### Author Response · Authors · 2025-11-26
> > > > > **Thank you**
> > > > >
> > > > > Thank you very much for your follow-up! We sincerely appreciate your feedback and the helpful suggestions that improved our work! All newly added analyses and results have been incorporated into the revised manuscript, and we are grateful for your contribution to enhancing the quality of the paper.

---

> ### Author Response · Authors · 2025-11-21
> **Response to Weakness 3 and Question 1**
>
> > **Weakness 3:** The theory part is interesting but might not be entirely novel. The fact that optimizing recommendation loss function is connected to the ranking metric NCDG [3] is relatively known in the community, and in my opinion the majority of knowledge distillation loss derivation in this paper is centered around this.
>
> We agree that the relationship between conventional recommendation losses and NDCG has been studied and we have acknowledged it in the related work section. However, **knowledge distillation losses differ fundamentally from standard supervised objectives**. In KD, (1) the “ground truth’’—the teacher’s outputs—is **real-valued**, not binary as in prior CE–NDCG analyses, and (2) KD in large-scale recommender systems is **sampling-based**, whereas existing theory assumes computation over the **entire item set**, rendering them unusable in practice. These two differences invalidate the assumptions underlying previous results.
>
> To overcome these gaps, we (i) develop an extension of the CE–NDCG connection that properly handles **real-valued teacher scores** using normalization and Jensen-type arguments, and (ii) introduce a **subset-based NDCG** together with a **key condition** (Assumption 4.3) that guarantees alignment between the distillation loss and ranking quality under item sampling. This condition has not appeared in prior “full-item’’ analyses but is crucial for making theory applicable to real industrial scenarios.
>
> In summary, our theoretical contribution is not a repetition of known results. **We extend CE–NDCG connections to the KD regime** and **provide the first sampling-compatible framework** that both motivates our method and **points to a more realistic theoretical framework for future industrial recommender research.**
>
> > **Question 1:** Can the proposed knowledge distillation method be applied to multi-modal recommender systems or just recommender systems without modality features?
>
> Since our method only requires the teacher to provide its final prediction logits, it imposes no constraints on the teacher’s backbone. In principle, this makes our approach fully applicable to multi-modal recommender systems. In practice, **we have also validated this by conducting additional experiments on multi-modal recommendation tasks**. We select three multi-modal datasets, i.e., Netflix, TikTok, and Amazon Electronics, and utilize BM3 as the teacher. The results show that our method continues to outperform all baselines by a significant margin in the multi-modal setting. The experimental results are given below:
>
> |     T→S      |  Method  |  Netflix   |            |   Tiktok   |            | Electronics |            |
> | :----------: | :------: | :--------: | :--------: | :--------: | :--------: | :---------: | :--------: |
> |              |          | Recall@20  |  NDCG@20   | Recall@20  |  NDCG@20   |  Recall@20  |  NDCG@20   |
> |  **BM3→MF**  | Teacher  |   0.1913   |   0.0809   |   0.0707   |   0.0254   |   0.0361    |   0.0152   |
> |              | Student  |   0.1420   |   0.0498   |   0.0417   |   0.0153   |   0.0182    |   0.0073   |
> |              |   ---    |    ---     |    ---     |    ---     |    ---     |     ---     |    ---     |
> |              | HetComp  |   0.1783   |   0.0755   |   0.0659   |   0.0238   |   0.0320    |   0.0133   |
> |              | PromptMM |   0.1732   |   0.0728   |   0.0637   |   0.0221   |   0.0297    |   0.0120   |
> |              |  TARec   |   0.1792   |   0.0760   |   0.0656   |   0.0230   |   0.0333    |   0.0139   |
> |              |  RCE-KD  | **0.1884** | **0.0797** | **0.0702** | **0.0253** | **0.0347**  | **0.0146** |
> |              |          |            |            |            |            |             |            |
> | **BM3→LGCN** | Teacher  |   0.1913   |   0.0809   |   0.0707   |   0.0254   |   0.0361    |   0.0152   |
> |              | Student  |   0.1507   |   0.0537   |   0.0562   |   0.0189   |   0.0211    |   0.083    |
> |              |   ---    |    ---     |    ---     |    ---     |    ---     |     ---     |    ---     |
> |              | HetComp  |   0.1811   |   0.0747   |   0.0629   |   0.0220   |   0.0317    |   0.0129   |
> |              | PromptMM |   0.1779   |   0.0728   |   0.0603   |   0.0215   |   0.0323    |   0.0137   |
> |              |  TARec   |   0.1834   |   0.0753   |   0.0651   |   0.0235   |   0.0341    |   0.0144   |
> |              |  RCE-KD  | **0.1907** | **0.0804** | **0.0705** | **0.0255** | **0.0359**  | **0.0153** |
>
> The table above is **Table 12** in our update submission. For detailed experimental settings and results, please check **Appendix D.2**.

---

> > ### Comment · Reviewer_irWM · 2025-11-24
> >
> > Thank you for your response, which addresses my concern on additional baselines for comparison. However, I still think that the theory part is a simple extension of a known conclusion (optimizing recommendation loss function is connected to the ranking metric NCDG) applied to knowledge distillation for recommendation. The resulting knowledge distillation loss function is adapted to consider only a subset of items, and is based on many strong assumptions, some of which may or may not hold in real-world recommendation settings.

---

> ### Author Response · Authors · 2025-11-25
> **Response to Concern on the Theory Part**
>
> We appreciate the reviewer’s continued attention to our theoretical section. However, we respectfully disagree with the reviewer’s conclusions on both aspects.
>
> > **(1) Regarding the claim that our theory is merely a “simple extension”:**
>
> Our analysis is nontrivial for three reasons.
>
> First, extending CE–NDCG connections to KD requires handling **real-valued teacher outputs**, which **necessitates new normalization and Jensen-type arguments** absent from prior binary-label CE–NDCG results.
>
> Second, we must work in a **sampling-based KD setting**, whereas all classical CE–NDCG analyses assume full-item losses and exact normalization. **Modeling this realistic setting (i.e., the definition of Partial NDCG), and precisely characterizing what NDCG-related guarantees can still be preserved under sampling (i.e., Theorem 4.4)**, is mathematically challenging and has not been addressed before.
>
> Third, our goal is to **retain as much of the CE–NDCG relationship as possible while introducing the fewest possible assumptions**. Crucially, **all definitions, assumptions, and conclusions in our framework are formulated exclusively over the sampled subsets**, not over the full item universe. This greatly complicates the analysis, yet is essential for relevance to real-world KD.
>
> For these reasons, our theory is not a direct or simple extension of existing CE–NDCG results, but **a new treatment tailored to real-world recommendation settings**.
>
> > **(2) Regarding the concern that the theory “cannot be used in real-world settings due to strong assumptions”:**
>
>  We introduce **only a single assumption—Assumption 4.3**. In Section 4.3, we explicitly acknowledge that it cannot be perfectly satisfied in practice. However, the purpose of our theory is not to claim that ideal conditions hold, but to **identify exactly what property is required** for KD losses to retain NDCG consistency under sampling, and to clarify **what benefits can be expected** if this property is met (Theorem 4.4).
>
> Crucially, **our entire method section is devoted to approximating Assumption 4.3 in practice**: $\mathcal L_1$ exploits the theoretical guarantee,$\mathcal L_2$ expands the subset toward making the assumption “more true,” and the $\gamma$-schedule enables the model to benefit from the theoretical insights even when the assumption cannot be strictly satisfied. In other words, **our theory reveals the core difficulty of real-world KD under sampling, and our method is designed precisely to overcome this difficulty.**
>
> Therefore, the assumption is not a flaw but a **diagnostic tool** that exposes what must be addressed. Our method operationalizes this insight and brings theoretical benefits into practice.

---

### Official Review · Reviewer_TXd5 · 2025-10-27

**Soundness:** 3
**Presentation:** 4
**Contribution:** 3
**Rating:** 8
**Confidence:** 5

**Summary:**

The paper is about Knowledge Distillation (KD) of Recommender Systems (RS).\
Cross-Entropy (CE) loss is the most widely adopted loss function for KD in other domains, however, it underperforms in RS (Figure 1).\
The author shows that it is because we need to use all items for CE, to align the KD with NDCG.\
However, in practice, we cannot use all items for CE, we need to sample some items for the computational burden.\
To this end, they propose to use top-ranked items both from the teacher and the student.

**Strengths:**

1. The motivation is sound
- KD is an important task for RS, since the model size increases with the number of increasing users and items.
- Also, CE is the most widely adopted loss function for KD in other domains.

2. The proposed method is well-formulated
- I like Section 4, where the authors show that full CE is the proper way to align KD with NDCG.
- Also, we know that full-item CE is infeasible in practice. So they also present partial-item CE and some conditions for that.
- Lastly,

3. The paper demonstrates the superiority of the proposed method with experiments on real-world datasets.

**Weaknesses:**

1. Additional experiment result
- A comparison including\
(1) vanilla CE\
(2) full-item CE\
(3) partial-item CE (top-ranked from Teacher)\
(4) partial-item CE (top-ranked from Student)\
(5) partial-item CE (top-ranked from Teacher or Student, i.e., the union of them)\
(6) RCE-KD\
would be beneficial to clearly present the motivation/superiority of the proposed method.

2. Minor comments
- Figure 1 needs to include RCE-KD, to show that vanilla CE actually can be improved with a simple modification.
- Citation format should be revised (\cite{} vs \citep{}).\
For example, in the first sentence of Introduction, "Recently, with the scaling law in recommender systems **Zhai et al. (2024)** being gradually discovered"\
It should be "Recently, with the scaling law in recommender systems **(Zhai et al., 2024)** being gradually discovered"
- Typo in line 275: "The first subset is the **interaction** between Q^T_u and Q^S_u" -> **intersection**

**Questions:**

Please refer to Weaknesses.

---

> ### Author Response · Authors · 2025-11-21
>
> Thank you very much for your recognition of our work!
>
> > **Weakness 1:** Additional experiment result
> >
> > - A comparison including
> >   (1) vanilla CE
> >   (2) full-item CE
> >   (3) partial-item CE (top-ranked from Teacher)
> >   (4) partial-item CE (top-ranked from Student)
> >   (5) partial-item CE (top-ranked from Teacher or Student, i.e., the union of them)
> >   (6) RCE-KD
> >   would be beneficial to clearly present the motivation/superiority of the proposed method.
>
> Thank you very much for your suggestion. We have incorporated several of the experiments you proposed in our ablation study (Section 6.4). Specifically, we compared (3) partial-item CE using the teacher’s top-ranked items (denoted as **RCE-KD w/o S** in our experiments), (4) partial-item CE using the student’s top-ranked items (**RCE-KD w/o T**), (5) partial-item CE using the top-ranked items from either teacher or student (**RCE-KD w/o sep**), and (6) the full **RCE-KD** method. Since in recommendation settings (1) vanilla CE is effectively equivalent to (3) partial-item CE (top-ranked from teacher), the only missing variant was (2) full-item CE. In the revised Figures 3 and 6, **we have now included the results for (2) full-item CE, denoted as full-CE in our paper.**
>
> As shown, our method performs even slightly above full-item CE in most cases, due to a tighter bound on NDCG than full-item CE (Xu et al., 2024). At the same time, it offers a substantial and unmatched improvement in training efficiency.
>
> Please check **Figure3, Figure 6, Section 6.4, and Appendix C.3** in our updated submission.
>
> > **Weakness 2:** Minor comments
> >
> > - Figure 1 needs to include RCE-KD, to show that vanilla CE actually can be improved with a simple modification.
> > - Citation format should be revised (\cite{} vs \citep{}).
> >   For example, in the first sentence of Introduction, "Recently, with the scaling law in recommender systems **Zhai et al. (2024)** being gradually discovered"
> >   It should be "Recently, with the scaling law in recommender systems **(Zhai et al., 2024)** being gradually discovered"
> > - Typo in line 275: "The first subset is the **interaction** between Q^T_u and Q^S_u" -> **intersection**
>
> Thank you very much for pointing out these errors. Correcting them has made our work more rigorous. We have revised the manuscript according to your comments and we have included RCE-KD in Figure 1. Please check **Figure 1, line 281, and all locations where citations are used** in our updated submission.

---

> > ### Comment · Reviewer_TXd5 · 2025-11-24
> >
> > I appreciate your response.\
> > I will keep my score.

---

> > > ### Author Response · Authors · 2025-11-25
> > > **Thank you very much for your affirmation!**
> > >
> > > Thank you very much for your affirmation!

---

### Official Review · Reviewer_PnSg · 2025-10-29

**Soundness:** 2
**Presentation:** 2
**Contribution:** 2
**Rating:** 4
**Confidence:** 3

**Summary:**

This paper studies why Cross-Entropy (CE) loss performs poorly in knowledge distillation for recommender systems. The authors prove CE loss only works well when the item set includes the student model's top-ranked items, but this conflicts with the goal of distilling the teacher's top items. They propose RCE-KD, which splits the teacher's top items into two groups and applies different sampling strategies. Experiments show their method consistently outperforms existing distillation approaches across multiple datasets and model configurations.

**Strengths:**

1. The paper provides rigorous theoretical analysis by establishing the connection between CE loss and partial NDCG in recommender system KD, identifying the critical closure assumption that determines when CE loss effectively bounds ranking performance.
2. The paper is well-structured and easy to follow.
3. This paper presents experimental results on multimodal recommendation scenarios, demonstrating their effectiveness.

**Weaknesses:**

1. The motivation for model compression in recommendation systems is questionable. Unlike CV/NLP models that require edge deployment, recommendation systems typically operate on cloud infrastructure where model size is rarely a bottleneck. The authors fail to provide compelling evidence that model size poses a significant problem in recommendation scenarios. Without clear justification for why compression is necessary in this cloud-based deployment context, the practical value of this work remains unconvincing.
2. The teacher models used in this study are relatively modest in size, with the largest containing fewer than 300M parameters. This contradicts the fundamental premise of knowledge distillation, which typically involves compressing truly large, unwieldy models like LLMs or large vision transformers. With teacher models of this scale, standard optimization and deployment techniques should suffice, making the proposed distillation approach potentially unnecessary for the problem at hand.
3. Current trends in recommendation system distillation focus on using large multimodal models as teachers to guide smaller ID-based models—a more practical scenario given the genuine computational overhead of multimodal architectures. However, this paper exclusively employs ID-only models for both teacher and student, which represents a less compelling use case. The authors should justify this choice or adopt the more realistic multimodal-to-ID distillation setting.
4. Recent work [1] has demonstrated the effectiveness of Wasserstein distance for knowledge distillation in recommendation systems. However, the authors do not compare their approach against alternative distance metrics, including Wasserstein-based methods.


[1] Zhuang, Ziyi, et al. "Bridging the Gap: Teacher-Assisted Wasserstein Knowledge Distillation for Efficient Multi-Modal Recommendation." Proceedings of the ACM on Web Conference 2025. 2025.

**Questions:**

1. Why were these three datasets chosen? The Amazon dataset is a widely-used benchmark in recommendation research but is absent here. Could the authors justify this choice or include Amazon datasets in the evaluation?
2. Why are standard sequential recommendation models like SASRec and GRU4Rec not included as baselines? These are established benchmarks, and their absence makes it difficult to assess the competitiveness of the proposed approach.

---

> ### Author Response · Authors · 2025-11-21
> **Reponse to Weakness 1 & 2**
>
> Thank you for reviewing our paper!
>
> > **Weakness 1:** The motivation for model compression in recommendation systems is questionable.
>
> We respectfully disagree that model compression is unmotivated for recommender systems simply because they run in the cloud. Large-scale industrial recommenders are *severely* constrained by memory, latency and serving cost, primarily due to massive embedding tables and high request volumes. For example, Mem-Rec reports that modern DLRM-style models use categorical features with tens of millions of values and embedding tables on the order of *hundreds of GB*, and shows that compressing these tables by up to **2900×** can keep AUC unchanged while improving embedding latency by up to **3.4×** [1]. Similarly, SHARK targets terabyte-scale embedding layers in production recommenders at Kuaishou and demonstrates that compression can save **70%** storage, free “thousands of machines”, and improve QPS by **30%** without online performance loss[2].
>
> Latency and throughput are also first-class concerns in cloud deployment: DeepLight accelerates CTR prediction by **46×** on Criteo and **27×** on Avazu with no degradation in accuracy, specifically to make deep recommenders deployable in ad serving[3]. NVIDIA’s best-practice guidelines for large-scale recommender systems likewise emphasize GPU memory limits and inference optimization as key deployment bottlenecks, not afterthoughts[4].
>
> Within this ecosystem, knowledge-distillation–based compression for ranking models is already recognized as valuable[5,6].  Our work follows this line: it targets the same real-world constraints (memory, latency, and serving cost) while maintaining or improving NDCG, and is therefore practically relevant even in cloud-based deployment settings.
>
> [1] "Mem-rec: Memory efficient recommendation system using alternative". ACML 2024.
>
> [2] "SHARK: A Lightweight Model Compression Approach for Large-scale Recommender Systems". CIKM 2023.
>
> [3] "DeepLight: Deep Lightweight Feature Interactions for Accelerating CTR Predictions in Ad Serving". WSDM 2021.
>
> [4] "Best Practices for Building and Deploying Recommender Systems". NVIDIA Developer Documentation.
>
> [5] "Distillation from Heterogeneous Models for Top-K Recommendation". WWW 2023.
>
> [6] "Continual Collaborative Distillation for Recommender System". KDD 2024.
>
> > **Weakness 2:** The teacher models used in this study are relatively modest in size, with the largest containing fewer than 300M parameters.
>
> We agree that the teacher models in our experiments are modest. This was deliberate so we could rely on public academic datasets and compare fairly with prior work. Still, our study includes the academic HSTU variant—a direct descendant of the industrial architecture proposed by Meta, whose production-scale deployment at Meta can reach 1.5 trillion parameters and serve billions of users. Although we cannot access that industrial-scale setup, our consistent gains on the academic HSTU baseline, combined with HSTU’s prevalence in real systems, give us strong reason to expect the same distillation strategy to remain effective when both the model and dataset sizes grow by orders of magnitude.

---

> ### Author Response · Authors · 2025-11-21
> **Reponse to Weakness 3**
>
> > **Weakness 3:** The authors should justify this choice or adopt the more realistic multimodal-to-ID distillation setting.
>
> Although recent studies explore multimodal-to-ID distillation, **ID-based models still dominate industrial-scale recommendation systems**. As noted in *ID-centric Pre-training for Recommendation*, “ID-based SR models still dominate most real-world systems,” and surveys such as *Where to Go Next for Recommender Systems* conclude that ID-based models have been the backbone of practical recommenders for over a decade. Moreover, in many production pipelines, multimodal signals are **offline-processed features**, while the final ranking model remains ID-centric. Thus, compressing large ID-based models is still of substantial practical value, especially as these models continue to grow in depth, width, and embedding size across system iterations.
>
> At the same time, our method is **backbone-agnostic** and not limited to ID-to-ID distillation. To demonstrate this, **we added experiments on three multimodal datasets**, where the **multimodal teacher** (BM3) distill into **ID-based students** (MF, LightGCN). Our approach remains effective and **significantly outperforms** all baselines in this multimodal-to-ID setting. It demonstrates that our method remains fully applicable to multimodal recommendation as well. The experimental results are given below:
>
> |     T→S      |  Method  |  Netflix   |            |   Tiktok   |            | Electronics |            |
> | :----------: | :------: | :--------: | :--------: | :--------: | :--------: | :---------: | :--------: |
> |              |          | Recall@20  |  NDCG@20   | Recall@20  |  NDCG@20   |  Recall@20  |  NDCG@20   |
> |  **BM3→MF**  | Teacher  |   0.1913   |   0.0809   |   0.0707   |   0.0254   |   0.0361    |   0.0152   |
> |              | Student  |   0.1420   |   0.0498   |   0.0417   |   0.0153   |   0.0182    |   0.0073   |
> |              |   ---    |    ---     |    ---     |    ---     |    ---     |     ---     |    ---     |
> |              | HetComp  |   0.1783   |   0.0755   |   0.0659   |   0.0238   |   0.0320    |   0.0133   |
> |              | PromptMM |   0.1732   |   0.0728   |   0.0637   |   0.0221   |   0.0297    |   0.0120   |
> |              |  TARec   |   0.1792   |   0.0760   |   0.0656   |   0.0230   |   0.0333    |   0.0139   |
> |              |  RCE-KD  | **0.1884** | **0.0797** | **0.0702** | **0.0253** | **0.0347**  | **0.0146** |
> |              |          |            |            |            |            |             |            |
> | **BM3→LGCN** | Teacher  |   0.1913   |   0.0809   |   0.0707   |   0.0254   |   0.0361    |   0.0152   |
> |              | Student  |   0.1507   |   0.0537   |   0.0562   |   0.0189   |   0.0211    |   0.083    |
> |              |   ---    |    ---     |    ---     |    ---     |    ---     |     ---     |    ---     |
> |              | HetComp  |   0.1811   |   0.0747   |   0.0629   |   0.0220   |   0.0317    |   0.0129   |
> |              | PromptMM |   0.1779   |   0.0728   |   0.0603   |   0.0215   |   0.0323    |   0.0137   |
> |              |  TARec   |   0.1834   |   0.0753   |   0.0651   |   0.0235   |   0.0341    |   0.0144   |
> |              |  RCE-KD  | **0.1907** | **0.0804** | **0.0705** | **0.0255** | **0.0359**  | **0.0153** |
>
> The table above is **Table 15** in our update submission. For detailed experimental settings and results, please check **Appendix D.2**.

---

> ### Author Response · Authors · 2025-11-21
> **Response to Weakness 4 and Question 1 & 2**
>
> > **Weakness 4:** the authors do not compare their approach against alternative distance metrics.
>
> The referenced work focuses specifically on multi-modal recommendation, which is a different problem setting from ours. For this reason, we initially did not include it in our comparisons. We apologize for the oversight. After carefully re-examining the method, we found that its Wasserstein-based distillation component can indeed be adapted to our scenario.
>
> Accordingly, **we have added this method as a baseline in our experiments,** **including the main experiments, the sequential recommendation and the newly added multimodal recommendation experiments**. Since the referenced method combines both logit- and feature-based distillation, whereas other baselines are purely logit-based, we remove the feature-based component from it to ensure a fair comparison. The experimental results show that our method consistently outperforms this approach. The experimental results in top-N recommendation in our main text is given below:
>
> | T→S           | Method | CiteULike  |            | Gowalla    |            | Yelp       |            |
> | ------------- | ------ | ---------- | ---------- | ---------- | ---------- | ---------- | ---------- |
> |               |        | R@20       | N@20       | R@20       | N@20       | R@20       | N@20       |
> | **MF→MF**     | TARec  | 0.0380     | 0.0171     | 0.1416     | 0.0993     | 0.0635     | 0.0327     |
> |               | RCE-KD | **0.0431** | **0.0194** | **0.1525** | **0.1047** | **0.0667** | **0.0345** |
> | **LGCN→LGCN** | TARec  | 0.0350     | 0.0157     | 0.1630     | 0.1112     | 0.0694     | 0.0352     |
> |               | RCE-KD | **0.0377** | **0.0171** | **0.1681** | **0.1163** | **0.0716** | **0.0369** |
> | **HSTU→HSTU** | TARec  | 0.0571     | 0.0299     | 0.1537     | 0.1005     | 0.0713     | 0.0369     |
> |               | RCE-KD | **0.0670** | **0.0366** | **0.1594** | **0.1058** | **0.0754** | **0.0400** |
>
> For complete results, please refer to **Table 1** in our updated submission. For results in sequential recommendation, please refer to **Table 12**. For results in multi-modal recommendation, please refer to **Table 15**.
>
> > **Question 1:** Could the authors justify this choice or include Amazon datasets in the evaluation?
>
> We selected these three datasets by following standard practice in prior distillation studies for recommendation systems (e.g., Lee et al., 2024; Kang et al., 2022; Kweon et al., 2021; Lee et al., 2019), where they are among the most commonly used benchmarks, ensuring fair comparison with existing work. We apologize for not including the Amazon dataset initially. To address this, we have added experiments on **Amazon Electronics** in our newly introduced multimodal setting, together with two new datasets: **Netflix and Tiktok**. The results show that our method continues to outperform all baselines by a significant margin in all datasets. The results are given in our response to Weakness 3. For detailed experimental settings and results, please check **Appendix D.2**.
>
> > **Question 2:** Why are standard sequential recommendation models like SASRec and GRU4Rec not included as baselines?
>
> Thank you for the question. In the main paper, we study a top-N recommendation setting, so models like SASRec and GRU4Rec were not included.
>
> Nevertheless, we appreciate the concern. In the sequential recommendation setting in Appendix D.1, we added a setting where both teacher and student are **SASRec**, and our method still shows **clear improvements** over all baselines. The results are given below:
>
> | T→S               | Method  | CiteULike  |            | Gowalla    |            | Yelp       |            |
> | ----------------- | ------- | ---------- | ---------- | ---------- | ---------- | ---------- | ---------- |
> |                   |         | Recall@10  | NDCG@10    | Recall@10  | NDCG@10    | Recall@10  | NDCG@10    |
> | **SASRec→SASRec** | Teacher | 0.0091     | 0.0067     | 0.0325     | 0.0214     | 0.0159     | 0.0087     |
> |                   | Student | 0.0054     | 0.0042     | 0.0160     | 0.0086     | 0.0100     | 0.0067     |
> |                   | ---     | ---        | ---        | ---        | ---        | ---        | ---        |
> |                   | CD      | 0.0073     | 0.0055     | 0.0250     | 0.0169     | 0.0133     | 0.0070     |
> |                   | RRD     | 0.0079     | 0.0057     | 0.0241     | 0.0165     | 0.0129     | 0.0069     |
> |                   | DCD     | 0.0083     | 0.0064     | 0.0269     | 0.0170     | 0.0143     | 0.0074     |
> |                   | HetComp | 0.0086     | 0.0066     | 0.0281     | 0.0174     | 0.0136     | 0.0070     |
> |                   | TARec   | 0.0080     | 0.0059     | 0.0271     | 0.0171     | 0.0130     | 0.0067     |
> |                   | RCE-KD  | **0.0092** | **0.0070** | **0.0303** | **0.0181** | **0.0154** | **0.0078** |
>
> The table above is in **Table 12** in our update submission.

---

> > ### Comment · Reviewer_PnSg · 2025-11-25
> >
> > I appreciate the detailed response. As most of my concerns are now resolved, I highly suggest that the authors include the new results in the revision. I will increase my scores accordingly.

---

> > > ### Author Response · Authors · 2025-11-26
> > > **Thank you**
> > >
> > > Thank you very much for your thoughtful follow-up! We truly appreciate your constructive feedback throughout the discussion process! We have incorporated all newly added experiments into the revised manuscript, and we warmly welcome you to review them.

---

### Official Review · Reviewer_xgs9 · 2025-11-01

**Soundness:** 2
**Presentation:** 3
**Contribution:** 2
**Rating:** 4
**Confidence:** 4

**Summary:**

This paper studies the cross-entropy (CE) loss in the field of recommendation knowledge distillation, and proposes a new adaptively-weighted distillation loss.

While the topic is important and the paper presents a method that is conceptually reasonable, I have some concerns on the unclear motivation and technical contribution.  As a result, I initially assign a score of 4, but I could raise my score if my concerns have been well addressed.

**Strengths:**

This work has the following strengths:

1.	This work studies on an important problem.

2.	The proposed method appears to be reasonable.

3.	Extensive experiments on real-world datasets have been conducted to verify the efficacy of the proposed method.

**Weaknesses:**

However, this work also has some limitations:

1.	My major concerns lie on the motivations. 1) Why is optimizing NDCG considered critical in the context of knowledge distillation? While NDCG is a standard metric for evaluating recommendation performance, it may not be an ideal loss function for distillation purposes. 2) Observation 4.5 is interesting, but it is derived solely from the CiteULike dataset, making the evidence less robust and limiting generalizability. 3) Assumption 4.3 imposes a relatively strong condition that may be difficult to satisfy. Is this assumption strictly necessary to establish the theoretical connection between CE and NDCG? Could weaker or more relaxed conditions be used instead? 4) There appears to be a gap between the theoretical analysis and the proposed method. Does the proposed method strictly satisfy Assumption 4.3? Can the proposed loss function tightly bound the NDCG metric as claimed?


2.	My another concerns lie on the technical contribution: 1) The theoretical relationship between NDCG and CE loss has been explored in prior work (e.g., [a1]). Connecting NDCG and CE in the specific setting of knowledge distillation may not constitute a highly novel contribution unless unique theoretical challenges are clearly articulated. What are the distinctive theoretical hurdles addressed in this work, and what new insights are provided? 2) The proposed method appears to be a relatively simple distillation trick. While simplicity is not inherently undesirable and sometimes could be a good properties, a deeper theoretical justification is needed to explain why this strategy is particularly effective or elegant for this task. What are the inherent merits or underlying principles that make it well-suited here?

3.	I also have some minor concerns on the experiments: 1) The baselines are relatively dated (before 2023). Incorporating more recent state-of-the-art methods would strengthen the evaluation. 2) The experiments are conducted only on three datasets. Including more datasets would better.

[a1] On the effectiveness of sampled softmax loss for item recommendation.

**Questions:**

Please refer to weaknesses.

---

> ### Author Response · Authors · 2025-11-21
> **Response to Weakness 1**
>
> Thank you for reviewing our paper!
>
> > **Weakness 1.1:** Why is optimizing NDCG considered critical in the context of knowledge distillation? While NDCG is a standard metric for evaluating recommendation performance, it may not be an ideal loss function for distillation purposes.
>
> Optimizing NDCG is essential in our distillation setting because **our goal is to transfer the teacher’s ranking ability**, which directly determines recommendation quality. As shown in prior work (e.g., DE-RRD, HetComp), distillation is most effective when the student inherits the teacher’s listwise ranking behavior rather than its raw scores. Since NDCG is the standard metric for measuring such ranking quality, it remains equally suitable for guiding distillation.
>
> Note that, **in our case, NDCG is computed with respect to the teacher rather than ground truth**, allowing the student to approximate the teacher’s ranking structure.
>
> Importantly, common KL-based distillation losses also implicitly optimize NDCG[a1], because under gradient descent, KL divergence reduces to cross-entropy.
>
> [a1] Hinton G, Vinyals O, Dean J. Distilling the knowledge in a neural network[J]. arXiv preprint arXiv:1503.02531, 2015.
>
> > **Weakness 1.2:** Observation 4.5 is interesting, but it is derived solely from the CiteULike dataset, making the evidence less robust and limiting generalizability.
>
> Thank you very much for your affirmation. Due to space constraints, we have included the complete experimental results in the appendix in our original submission. **Figure 4 in Appendix A.1** presents experimental results across **all three datasets** (CiteULike, Gowalla, Yelp) and **all backbones** (MF, LightGCN, HSTU). While showing slight variations, these results consistently demonstrate the observation described in our main text.
>
> > **Weakness 1.3:** Assumption 4.3 imposes a relatively strong condition that may be difficult to satisfy. Is this assumption strictly necessary to establish the theoretical connection between CE and NDCG? Could weaker or more relaxed conditions be used instead?
>
> Unfortunately, we were unable to find a more lenient alternative that would serve the same purpose. As demonstrated in the proof of Theorem 4.4 (Eq. 25), to compute NDCG precisely, we must know the exact ranking of the item. To obtain the precise ranking, we must traverse all items with scores higher than that of the current item. And it is precisely this that makes Assumption 4.3 indispensable, as it ensures that all higher-ranked items are included in our subset of items. Of course, we could employ more relaxed assumptions by relaxing the bound, but this risks triggering a large number of ineffective optimizations due to the loosened boundary.
>
> > **Weakness 1.4:** There appears to be a gap between the theoretical analysis and the proposed method. Does the proposed method strictly satisfy Assumption 4.3? Can the proposed loss function tightly bound the NDCG metric as claimed?
>
> Since our method approximates Assumption 4.3 through sampling with a complex distribution, we regret that we cannot guarantee strict satisfaction. However, **Appendix C.2** in our original submission has presented the efficiency of our sampling strategy for approximating Assumption 4.3. As can be seen, **the discrepancy between our adjusted item set $\mathcal A_u$ and the ideal item set that strictly satisfies Assumption 4.3 is negligible**. Concretely, **after training for only 20% of the total number of epochs, the set we constructed can approximate the ideal set** (which strictly satisfies Assumption 4.3) **with a 95% overlap rate**. This indicates that our method can quickly bring $(\mathcal Q\_u^T)\_2$ to approximate Assumption 4.3.
>
> To verify that our loss function indeed constrains NDCG more tightly, we follow the evaluation protocol widely adopted in prior work on surrogate ranking losses: we visualize the evolution of training NDCG across epochs. The corresponding results appear in **Appendix A.2**. They consistently show that, relative to the vanilla CE baseline, **our method delivers higher training NDCG, a steeper growth trajectory, and earlier convergence**. These observations jointly indicate that our improved loss keeps NDCG closer to its optimal bound.

---

> ### Author Response · Authors · 2025-11-21
> **Response to Weakness 2**
>
> > **Weakness 2.1:** What are the distinctive theoretical hurdles addressed in this work, and what new insights are provided?
>
> While prior work has connected CE and NDCG, those analyses rely on assumptions that do **not** hold in the knowledge distillation setting.
>
> First, earlier results assume **binary relevance (0/1)**, whereas in KD the teacher outputs **real-valued scores**. This requires nontrivial adaptations—such as appropriate normalization and the use of Jensen’s inequality—to formally relate the distillation loss to NDCG.
>
> Second, **previous theory assumes optimization over the entire item universe**, but large-scale recommenders rely on **sampling-based training**, where only a subset of items is accessible at each step. This makes full-list theoretical conclusions inapplicable. **Modeling this realistic setting (i.e., the definition of Partial NDCG), and precisely characterizing what NDCG-related guarantees can still be preserved under sampling (i.e., Theorem 4.4)**, is mathematically challenging and has not been addressed before.
>
> Third, **all definitions, assumptions, and conclusions in our framework are formulated exclusively over the sampled subsets**, and none pertain to the full item universe. This greatly complicates the analysis, yet is essential for relevance to real-world KD.
>
> In summary, our contributions are: (1) extending CE–NDCG analysis to the KD regime with **real-valued teacher outputs**, and (2) establishing the first **completely and exclusively sampling-aware theoretical framework** that both motivates our method and points to a more realistic theoretical framework for future industrial recommender research.
>
> > **Weakness 2.2:** The proposed method appears to be a relatively simple distillation trick. What are the inherent merits or underlying principles that make it well-suited here?
>
> Our method is explicitly guided by one design principle: **we aim to approximate Assumption 4.3 as closely as real recommender systems allow, while ensuring that even slight deviations still let the CE loss inherit the same theoretical advantages.**
>
> To operationalize this principle, we employ four coordinated components:
>
> 1. **Partitioning (Sec. 5.1).** We divide items into closure-compliant and non-compliant subsets according to whether Assumption 4.3 is satisfied, so each receives the treatment that best aligns with Assumption 4.3, rather than letting violations obscure the theory.
>
> 2. **Guarantee for compliant items (Sec. 5.2).** The compliant subset fully satisfies the assumption, so their losses strictly follow Theorem 4.4 and optimize the ranking objective with firm guarantees.
>
> 3. **Approximation for non-compliant items (Sec. 5.3).** We introduce a score-based sampler that steadily reshapes the violating subset so it nearly satisfies Assumption 4.3. In practice, roughly 20 % of training epochs (see Weakness 1.4 response) suffice for these items to completely inherit the benefits of Theorem 4.4.
>
> 4. **Adaptive fusion (Sec. 5.4).** A dynamic weighting scheme balances two subsets, suppressing the influence of residual violations while allowing approximation terms to supplement when contributions from strictly compliant terms are insufficient.
>
> Viewed together, what seems like a simple distillation tweak is in fact a mechanism for continuously steering real-world data toward the theoretical sweet spot and rejuvenating CE-loss guarantees.

---

> ### Author Response · Authors · 2025-11-21
> **Response to Weakness 3**
>
> > **Weakness 3.1:** Incorporating more recent state-of-the-art methods would strengthen the evaluation.
>
> Based on your proposal and that of another reviewer, we have incorporated TARec (WWW'25) as a new baseline. It employs the Teacher-Assistant as intermediate model and utilizes the Wasserstein Distance as metric. Moreover, we have added promptMM (WWW'24) as a new baseline in our newly added multimodal recommendation task. Our method consistently outperforms these baselines across all settings.
>
> The experimental results in top-N recommendation in our main text is given below:
>
> | T→S           | Method | CiteULike  |            | Gowalla    |            | Yelp       |            |
> | ------------- | ------ | ---------- | ---------- | ---------- | ---------- | ---------- | ---------- |
> |               |        | R@20       | N@20       | R@20       | N@20       | R@20       | N@20       |
> | **MF→MF**     | TARec  | 0.0380     | 0.0171     | 0.1416     | 0.0993     | 0.0635     | 0.0327     |
> |               | RCE-KD | **0.0431** | **0.0194** | **0.1525** | **0.1047** | **0.0667** | **0.0345** |
> | **LGCN→LGCN** | TARec  | 0.0350     | 0.0157     | 0.1630     | 0.1112     | 0.0694     | 0.0352     |
> |               | RCE-KD | **0.0377** | **0.0171** | **0.1681** | **0.1163** | **0.0716** | **0.0369** |
> | **HSTU→HSTU** | TARec  | 0.0571     | 0.0299     | 0.1537     | 0.1005     | 0.0713     | 0.0369     |
> |               | RCE-KD | **0.0670** | **0.0366** | **0.1594** | **0.1058** | **0.0754** | **0.0400** |
> | **HSTU→MF**   | TARec  | 0.0370     | 0.0173     | 0.1603     | 0.1100     | 0.0663     | 0.0350     |
> |               | RCE-KD | **0.0437** | **0.0197** | **0.1677** | **0.1163** | **0.0692** | **0.0360** |
> | **LGCN→MF**   | TARec  | 0.0538     | 0.0300     | 0.1527     | 0.1023     | 0.0762     | 0.0378     |
> |               | RCE-KD | **0.0595** | **0.0324** | **0.1624** | **0.1065** | **0.0805** | **0.0419** |
>
> For complete results, please refer to **Table 1** in our updated submission. For results in sequential recommendation, please refer to **Table 12**. For results in multi-modal recommendation, please refer to **Table 15**.
>
> > **Weakness 3.2:** Including more datasets would better.
>
> Incorporating feedback from you and the other reviewers, we have added a new recommendation scenario—**multimodal recommendation**—and conducted experiments on **three new datasets**: Netflix, Tiktok, and Amazon Electronics. Our approach remains effective and **significantly outperforms** all baselines in all these new datasets. The experimental results are given below:
>
> |     T→S      |  Method  |  Netflix   |            |   Tiktok   |            | Electronics |            |
> | :----------: | :------: | :--------: | :--------: | :--------: | :--------: | :---------: | :--------: |
> |              |          | Recall@20  |  NDCG@20   | Recall@20  |  NDCG@20   |  Recall@20  |  NDCG@20   |
> |  **BM3→MF**  | Teacher  |   0.1913   |   0.0809   |   0.0707   |   0.0254   |   0.0361    |   0.0152   |
> |              | Student  |   0.1420   |   0.0498   |   0.0417   |   0.0153   |   0.0182    |   0.0073   |
> |              |   ---    |    ---     |    ---     |    ---     |    ---     |     ---     |    ---     |
> |              | HetComp  |   0.1783   |   0.0755   |   0.0659   |   0.0238   |   0.0320    |   0.0133   |
> |              | PromptMM |   0.1732   |   0.0728   |   0.0637   |   0.0221   |   0.0297    |   0.0120   |
> |              |  TARec   |   0.1792   |   0.0760   |   0.0656   |   0.0230   |   0.0333    |   0.0139   |
> |              |  RCE-KD  | **0.1884** | **0.0797** | **0.0702** | **0.0253** | **0.0347**  | **0.0146** |
> |              |          |            |            |            |            |             |            |
> | **BM3→LGCN** | Teacher  |   0.1913   |   0.0809   |   0.0707   |   0.0254   |   0.0361    |   0.0152   |
> |              | Student  |   0.1507   |   0.0537   |   0.0562   |   0.0189   |   0.0211    |   0.083    |
> |              |   ---    |    ---     |    ---     |    ---     |    ---     |     ---     |    ---     |
> |              | HetComp  |   0.1811   |   0.0747   |   0.0629   |   0.0220   |   0.0317    |   0.0129   |
> |              | PromptMM |   0.1779   |   0.0728   |   0.0603   |   0.0215   |   0.0323    |   0.0137   |
> |              |  TARec   |   0.1834   |   0.0753   |   0.0651   |   0.0235   |   0.0341    |   0.0144   |
> |              |  RCE-KD  | **0.1907** | **0.0804** | **0.0705** | **0.0255** | **0.0359**  | **0.0153** |
>
> The table above is **Table 15** in our update submission. For detailed experimental settings and results, please check **Appendix D.2**.

---

> ### Author Response · Authors · 2025-11-27
> **Grateful for your thorough review; we are glad to provide any additional clarification**
>
> Thank you again for your thoughtful and highly detailed review. We sincerely appreciate the care and rigor you invested in evaluating our work. If there are any remaining questions or points that you would like us to clarify further, we would be very glad to continue the discussion. Please feel free to let us know, as your feedback is greatly valued.

---

### Meta-Review · Area_Chair_EzYw · 2025-12-24

**Summary:**

This paper investigates the relationship between cross-entropy (CE) loss and NDCG loss for ranking-based distillation. The authors show
1) Full-list CE is a lower bound for NDCG loss, if the ground-truth scores have a certain form; and that
2) Partial-item CE may not be a lower bound, unless a critical "closure" condition is met.

Then, the authors proposes a CE variant, which contains two parts: the intersection of teacher's and student's top-N lists, and the remaining items in the teacher's top-N list. For the fist part, a traditional CE loss is applied, and the for second part, some reweighting sampling is applied when calculating the CE loss.

Strengths:
* The attempt to connect CE and NDCG in a theoretical manner is refreshing
* Empirical performance appears strong as such distilled student often matches, or even outperforms, the teacher.

Weaknesses:
I concur with Reviewers xgs9 and PnSg in that the theoretical significance and contributions are overly stated.

1. It's unclear why NDCG loss is theoretically superior than CE loss. In the rebuttal, the authors highlighted that they want to "transfer the teacher’s ranking ability". But this is an empty argument.

2. Full-list CE being a lower bound of NDCG depends on the ground-truth score having a certain form, induced from the teacher's logits. However, neural networks are usually not well calibrated (e.g., temperature and ranking are usually not ideal), so it's unclear how meaningful the result is.

3. Partial-list CE fails to be a lower bound NDCG unless a strict condition is met. However, it's unclear how important it is to remain optimizing a lower bound. The authors observed a challenge that the top-K lists of the teacher and student do not match, but it is common to do top-1 KD (i.e., hard KD). In fact, hard KD works particularly well for LLMs (containing tens of thousands of items), which contradicts with the observation made by the authors.

4. The proposed method is a heuristic, it's unclear whether the proposed method itself a lower-bound optimization (likely not).

I encourage the authors to rephrase their theoretical contributions and discuss them as a Limitation section.

**Reviewer Concerns:**

The reviewers raised a few major concerns, including motivations and lack of baselines.

**Reviewer Scores:**

The authors have provided additional experiments. However, the concerns about motivation and theoretical contributions cannot be easily addressed by rebuttal or revision.

---

### Decision · Program_Chairs · 2026-01-26

Accept (Poster)